# OmniAID: Decoupling Semantics and Artifacts for Universal AI-Generated Image Detection in the Wild

**Yuncheng Guo** [1]  **Junyan Ye** [2 1]  **Chenjue Zhang** [3]  **Hengrui Kang** [4 1]  **Haohuan Fu** [3]  **Conghui He** [1]  **Weijia Li** [3 1]

## Abstract

A truly universal AI-Generated Image (AIGI) detector must simultaneously generalize across diverse generative models and varied semantic content. Current methods learn a single, entangled forgery representation, conflating content-dependent flaws with content-agnostic artifacts, and are further constrained by outdated benchmarks. We propose **OmniAID**, a novel framework centered on a decoupled Mixture-of-Experts (MoE) architecture that separates: (1) semantic flaws across distinct content domains via Routable Specialized Semantic Experts, and (2) content-agnostic universal artifacts from content-dependent flaws via a Fixed Universal Artifact Expert. A two-stage training strategy first specializes experts independently with domain-specific hard-sampling, then trains a lightweight gating network for effective input routing. By explicitly decoupling "what is generated" (content-specific flaws) from "how it is generated" (universal artifacts), OmniAID achieves robust generalization. We also introduce **Mirage**, a large-scale, contemporary dataset comprising a modern training set and a challenging test set. Extensive experiments demonstrate that OmniAID surpasses existing detectors, establishing a new standard for AIGI detection against modern, in-the-wild threats. Code is available at https://github.com/yunncheng/OmniAID.

## 1. Introduction

The rapid proliferation of generative models, from Diffusion Models (DMs) to LLM-driven text-to-image technology

[1]Shanghai Artificial Intelligence Laboratory [2]Sun Yat-Sen University [3]Tsinghua Shenzhen International Graduate School, Tsinghua University [4]Shanghai Jiao Tong University. Correspondence to: Weijia Li <liweijia@sz.tsinghua.edu.cn>.

*Proceedings of the $43^{rd}$ International Conference on Machine Learning*, Seoul, South Korea. PMLR 306, 2026. Copyright 2026 by the author(s).

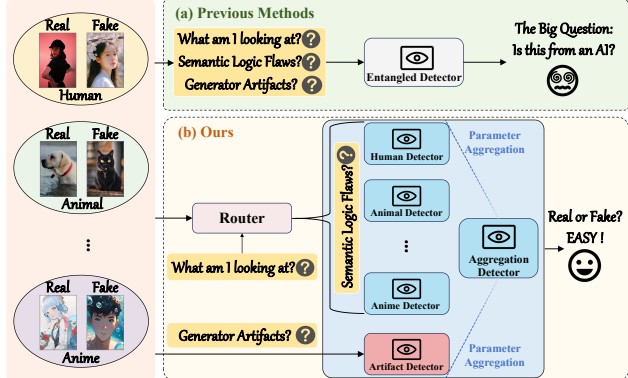

*Figure 1.* (a) Previous methods suffer from a monolithic, entangled representation, merging semantic flaws and universal artifacts, thereby restricting universality. (b) Our OmniAID solves this via decoupling: an input Router routes the image, specialized Semantic Detectors handle high-level flaws, and an Artifact Detector handles low-level features. The parameters from these active detectors are then aggregated into a final Aggregation Detector, which makes the robust, disentangled decision.

(Rombach et al., 2022; Black Forest Labs, 2024; Ye et al., 2025b; Yan et al., 2025c; Ye et al., 2025a), has saturated the digital ecosystem with highly photorealistic synthetic media. This trend renders the development of a truly universal AI-Generated Image (AIGI) detector a paramount challenge in digital forensics. Research in AIGI detection has broadly evolved along two lines: artifact-specific methods targeting low-level generator fingerprints (Wang et al., 2020; Qian et al., 2020; Jeong et al., 2022), and the now-dominant approach leveraging Vision Foundation Models (VFMs) (Radford et al., 2021; Oquab et al., 2023). While adapting pre-trained VFMs via Parameter-Efficient Fine-Tuning (PEFT) (Hu et al., 2022; Kong et al., 2023; Yan et al., 2025b) has substantially improved generalization, two bottlenecks still limit reliable deployment in open scenarios.

First, they learn a **monolithic and entangled representation**. Current state-of-the-art (SOTA) detectors merge all forgery clues into a single feature space. This entanglement, as illustrated in Figure 1 (a), proves problematic because it indiscriminately mixes high-level, content-dependent semantic flaws (e.g., distorted faces, impossible architecture) with low-level, content-agnostic universal artifacts (e.g.,

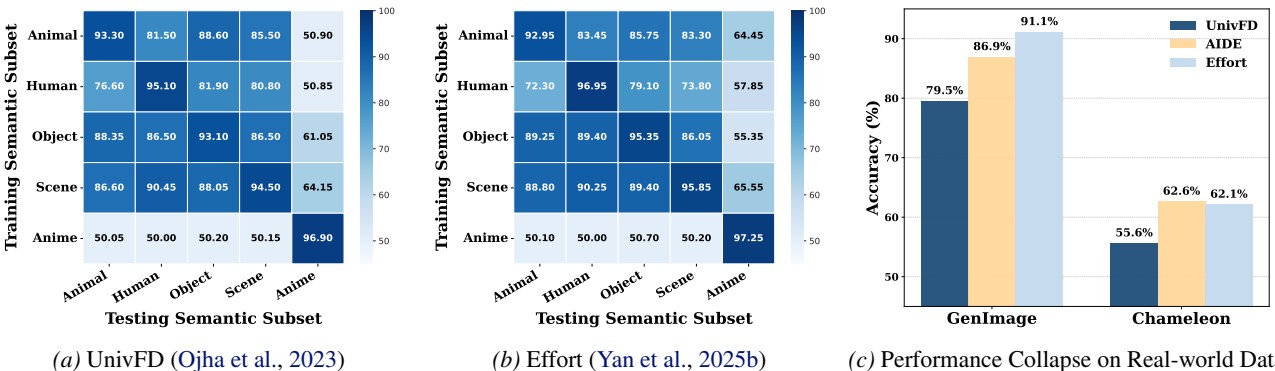

*(a)* UnivFD (Ojha et al., 2023)  *(b)* Effort (Yan et al., 2025b)  *(c)* Performance Collapse on Real-world Data

*Figure 2.* **Semantic Generalization Gaps and Benchmark Limitations.** (a)-(b) reveal poor cross-domain generalization on a **Mirage** subset, especially for the **Anime**, **Human**, and **Animal** domains. (c) illustrates how GenImage SDv1.4 (Zhu et al., 2023b)-trained models collapse on Chameleon (Yan et al., 2025a), underscoring profound benchmark limitations against in-the-wild distributional shift.

generator-specific frequency patterns), which in turn leads to practical failures: detectors trained on one semantic domain (e.g., Animal) exhibit poor generalization to others (e.g., Scene), as illustrated in Figures 2a and 2b. We posit that this failure stems from the VFM's core pre-training, which is not innately optimized to identify AIGI signals. Indeed, recent efforts have pursued different routes to mitigate this: some (Tan et al., 2025) reinforce semantic-level matching to improve category generalization, yet remain fragile under in-the-wild distributional shift; others (Chen et al., 2024; Rajan et al., 2024) construct hard negatives to compel artifact learning, yet overlook the fact that VFM-based representations remain semantically dominated, leaving the model's semantic capacity underutilized.

The second, equally critical challenge is **the crisis of outdated benchmarks**. Existing datasets (Zhong et al., 2023; Zhu et al., 2023b) are predominantly composed of images from older models (e.g., GANs (Goodfellow et al., 2014), early Stable Diffusion (Rombach et al., 2022)); consequently, detectors trained on them lack robustness to contemporary threats. As Figure 2c illustrates, SOTA methods trained on GenImage (Zhu et al., 2023b) perform well on its internal test set but fail significantly when evaluated on the more challenging, real-world Chameleon (Yan et al., 2025a) dataset. This stark performance gap reveals that existing academic leaderboards no longer reflect real-world robustness, mandating the development of new benchmarks that capture modern, real-world scenarios.

To address these twin bottlenecks, we propose **OmniAID**, a novel Mixture-of-Experts (MoE) architecture designed to explicitly decouple forgery traces. Our hybrid system features *Routable Specialized Semantic Experts* for content-specific flaws and one *Fixed Universal Artifact Expert* for content-agnostic fingerprints. This architecture is optimized via a bespoke two-stage training strategy: we first train the experts for specialization, then freeze them to train a

lightweight router. Concurrently, to address the "crisis of outdated benchmarks," we introduce **Mirage**, a new large-scale data foundation, including **Mirage-Train** for realistic model development and **Mirage-Test**, a challenging public test set built from held-out SOTA generators optimized for photorealism. By decoupling "what is generated" (semantics) from "how it is generated" (artifacts), OmniAID achieves a more robust, interpretable, and generalizable system, as confirmed by comprehensive validation on both traditional benchmarks and our new Mirage dataset. Our core contributions are:

1. We propose **OmniAID**, a novel MoE framework that dually decouples: (1) semantic flaws across distinct content domains via specialized *Routable Semantic Experts*, and (2) content-dependent flaws from content-agnostic artifacts via a *Fixed Universal Artifact Expert*.
2. We design a novel two-stage training strategy (expert specialization followed by router training) to efficiently optimize expert roles. This enables OmniAID to establish a new state-of-the-art in robust detection, surpassing prior monolithic detectors.
3. We contribute **Mirage**, a new large-scale data foundation comprising **Mirage-Train** (a modern training set) and **Mirage-Test** (a new, highly challenging public test set). This provides a rigorous and realistic evaluation against high-fidelity, real-world threats.

## 2. Related Work

The field of AI-generated image (AIGI) detection has evolved in lockstep with the rapid advancement of generative models, primarily bifurcating into two principal methodologies. While an emerging trend utilizes Large Multimodal Models (LMMs) for explainable detection (Ye et al., 2024; Wen et al., 2025; Kang et al., 2025; Xu et al., 2025b;a), this direction is beyond the scope of our work, which focuses on robust, generalizable detection via the aforementioned two

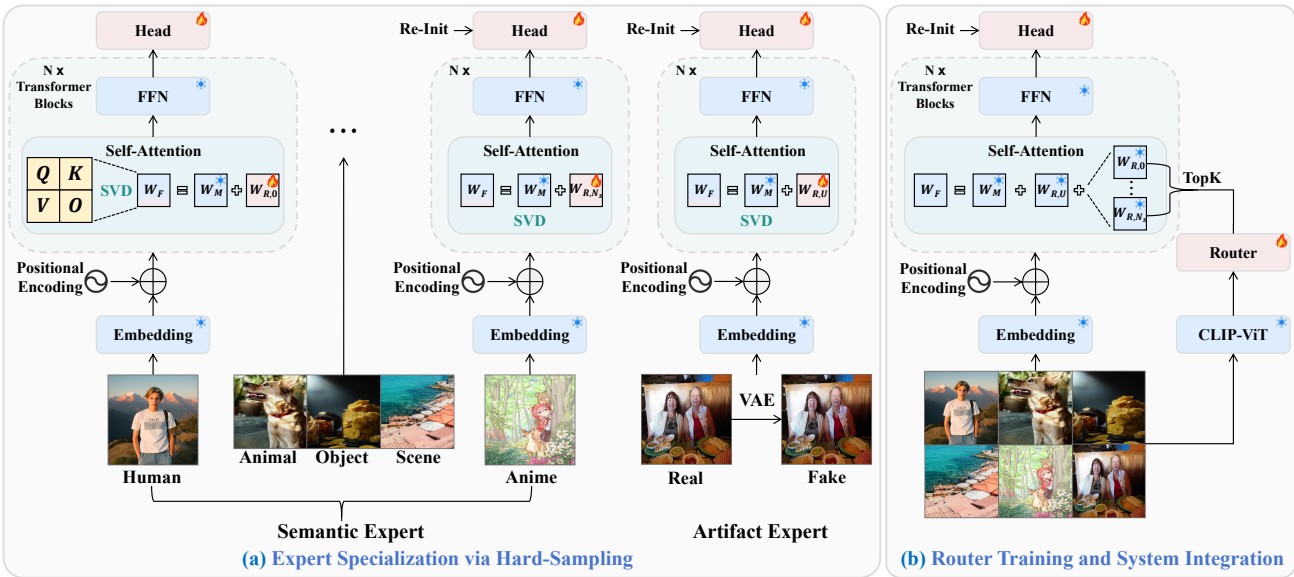

*Figure 3.* **Architectural overview of the proposed OmniAID framework**. The model employs a two-stage training strategy. **Stage 1 (a):** Expert Specialization, where domain-specific semantic experts (e.g., Human, Anime) and a universal Artifact Expert, both implemented as residual matrices after SVD decomposition, are trained independently using domain-specific data. **Stage 2 (b):** Router Training, where a lightweight router is trained, and the system integrates the weights from various experts into a final weight.

paradigms.

**Artifact-Specific Detection.** The first principal methodology centers on *fake pattern learning*, aiming to mine discriminative traces inherent to the generation process. These methods hypothesize that generative models leave unique, systematic fingerprints. For instance, initial studies demonstrated that standard CNNs, such as the ResNet (He et al., 2016) used in CNNSpot (Wang et al., 2020), could achieve strong detection performance on images from known generators. However, this approach is quickly found to overfit generator-specific patterns, exhibiting poor generalization to unseen generators. This limitation prompts subsequent research into more explicit artifact-mining techniques. Frequency-domain analyses (Qian et al., 2020; Jeong et al., 2022; Tan et al., 2024a) exploit spectral inconsistencies using high-pass filtering or frequency augmentation, whereas spatial-domain methods target pixel or texture statistics (Liu et al., 2021; Nataraj et al., 2019). Further studies leverage gradient information (Tan et al., 2023) or investigate generator-specific upsampling operations (Tan et al., 2024b). The primary limitation of this paradigm remains its brittleness: these techniques are often highly sensitive to generator architectures, noise, and compression, and thus struggle to generalize (Ojha et al., 2023).

**VFM-Based Generalizable Detection.** Addressing the generalization limits of artifact-specific detectors, a second, now-dominant paradigm leverages the rich, high-level representations of Vision Foundation Models (VFMs) such as CLIP (Radford et al., 2021) and DINOv2 (Oquab al.,

2023). UnivFD (Ojha et al., 2023) pioneers this by fine-tuning only a lightweight classification head. Subsequent works improve VFM detectors from different angles. PEFT-based detectors, such as LoRA adaptations (Hu et al., 2022; Kong et al., 2023) and the SVD-based Effort (Yan et al., 2025b), preserve the VFM semantic prior with lightweight updates. C2P-CLIP (Tan et al., 2025) goes further along this semantic-generalization route by injecting category-common prompts into CLIP to strengthen category-level similarity matching. Such semantic-oriented strategies are appealing, but their generalization can still degrade when test-time semantic distributions shift rapidly or fall outside the category regularities seen during training. AIDE (Yan et al., 2025a) represents a related hybrid-fusion strategy, showing that semantic/contextual features and low-level pixel-frequency evidence such as DCT-based cues can provide complementary signals. A different line constructs semantically aligned hard negatives, where real and reconstructed images share nearly identical content, to reduce the detector's reliance on content differences or dataset-specific semantic biases. For example, DRCT (Chen et al., 2024) and AlignedForensics (Rajan et al., 2024) use reconstruction-based counterparts to encourage detectors to focus on intrinsic generative traces. These methods improve artifact sensitivity, but their emphasis on artifact supervision can leave content-dependent semantic flaws underutilized. OmniAID aims to bridge these two directions by modeling semantic flaws and universal artifacts as complementary evidence, rather than relying solely on strengthened semantic matching or artifact-focused reconstruction supervision.

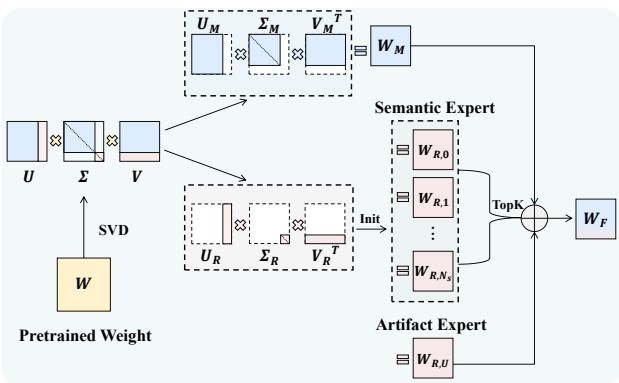

*Figure 4.* **SVD-based Weight Decomposition for Orthogonal MoE Adaptation.**

## 3. Methodology

We propose **OmniAID**, a universal AIGI detection framework (overviewed in Figure 3) designed to achieve a dual decoupling of forgery traces. It decouples (*i*) semantic flaws across distinct content domains and (*ii*) these content-dependent flaws from content-agnostic universal artifacts. To this end, we construct a content-aware multi-expert system with experts initialized within an orthogonal residual subspace (Yan et al., 2025b), enabling the explicit capture of diverse forgery fingerprints.

### 3.1. Hybrid Orthogonal MoE Architecture

Specifically, we decompose the weight matrix $\mathbf{W} \in \mathbb{R}^{d_{out} \times d_{in}}$ of a CLIP-ViT (Dosovitskiy, 2020; Radford et al., 2021) attention layer via SVD. Let $d = \min(d_{out}, d_{in})$. Given a rank $r$, we partition $\mathbf{W}$ into two orthogonal components, $\mathbf{W} = \mathbf{W}_M + \mathbf{W}_R$, defined as:

$$\mathbf{W}_M = \mathbf{U}_{:d-r}\mathbf{\Sigma}_{:d-r}\mathbf{V}_{:d-r}^T, \tag{1}$$

$$\mathbf{W}_R = \mathbf{U}_{d-r:}\mathbf{\Sigma}_{d-r:}\mathbf{V}_{d-r::}^T. \tag{2}$$

Here, $\mathbf{W}_M$ is the **frozen principal subspace** comprised of the top $d - r$ singular components, preserving the robust pre-trained knowledge of the base model. In contrast, $\mathbf{W}_R$ defines the **residual subspace** spanned by the remaining bottom $r$ components, which serves as the orthogonal basis for our adaptive components. As shown in Figure 4, Omni-AID instantiates a hybrid expert pool $\mathcal{E}$ within this residual manifold, enabling fine-grained decoupling of forgery traces via two specialized functional groups:

**(1) Specialized Semantic Experts.** A set of $N_S$ domain-specific experts $\mathcal{E}_S = \{e_1, e_2, \ldots, e_{N_S}\}$ is responsible for modeling the unique flaw patterns associated with distinct semantic domains (e.g., human faces, animals).

**(2) Universal Artifact Expert.** A single, universal expert $\mathcal{E}_U$ is designated to capture content-agnostic foren-

sic traces introduced by generation or reconstruction processes. Unlike semantic experts, $\mathcal{E}_U$ is not tied to a particular content domain; it is trained with semantically aligned real/reconstructed pairs so that image content is largely controlled and artifact traces become the primary discriminative signal. Since such traces can appear in any type of image, $\mathcal{E}_U$ remains active during every forward pass instead of competing with semantic experts in the router.

**Routing Mechanism.** A lightweight gating network $\mathcal{R}$ (implemented as an MLP) functions as a single global router, in contrast to traditional layer-specific routers, to select semantic experts. This global design is integral to our two-stage training strategy, facilitating model-wide specialization. To ensure stable, semantic-based routing, $\mathcal{R}$ operates on features from a separate, frozen CLIP-ViT encoder. During Stage 2 and inference, the router's selected top-$k_S$ semantic experts are combined with the universal expert $\mathcal{E}_U$ to form the active expert ensemble. Top-$k_S$ routing naturally supports mixed-category inputs by allowing multiple semantic experts to contribute jointly, while the always-active artifact expert provides category-agnostic evidence when the input semantics are ambiguous or partially outside the predefined taxonomy.

**Final Weight Composition.** As visualized in Figure 4, the final layer weight $\mathbf{W}_F$ is dynamically composed. It consists of the frozen principal subspace $\mathbf{W}_M$, the fixed Universal Artifact Expert ($\mathbf{W}_{R,U}$), and the weighted sum of the top-$k_S$ active semantic experts ($\mathbf{W}_{R,i}$). For a given input $\mathbf{x}$, the router $\mathcal{R}$ produces logits $\mathbf{z_x} \in \mathbb{R}^{N_S}$. Let $S$ be the set of top-$k_S$ indices selected by the router's gating weights $g_i = (\mathrm{Softmax}(\mathbf{z_x}))_i$. The final composed weight is:

$$\mathbf{W}_F = \mathbf{W}_M + \mathbf{W}_{R,U} + \sum_{i \in S} g_i \cdot \mathbf{W}_{R,i}. \tag{3}$$

### 3.2. Two-Stage Decoupled Training Strategy

The optimization of OmniAID is decoupled into two sequential stages to ensure both expert specialization and router accuracy, as illustrated in Figure 3.

#### 3.2.1. STAGE 1: EXPERT SPECIALIZATION VIA HARD-SAMPLING

In this stage, the router $\mathcal{R}$ and all experts, except for one, are frozen. A single target expert $e_i \in \mathcal{E}_S$ is activated and trained on an expert-specific hard-sampling set. Note that hard sampling here refers to evidence-targeted data construction rather than conventional hard example mining. Specifically, semantic experts are trained on their corresponding semantic domains to capture domain-specific flaws, while the artifact expert is trained on semantically aligned real/reconstructed pairs whose content is nearly invariant, making low-level reconstruction traces the main

discriminative signal. For stability and to ensure expert independence, we reinitialize the final classification head each time a new expert is trained. Only the low-rank residual components $\mathbf{U}_{d-r:}, \boldsymbol{\Sigma}_{d-r:}, \mathbf{V}_{d-r:}$ of the active expert and the classification head are trainable. The objective for the active expert $e_a$ is:

$$\mathcal{L}_{\text{Stage1}} = \mathcal{L}_{\text{cls}} + \lambda_1 \mathcal{L}_{\text{orth}}. \qquad (4)$$

Here, $\mathcal{L}_{\text{cls}}$ is the primary Cross-Entropy (CE) classification loss. To facilitate semantic decoupling, we adapt the orthogonality constraint $\mathcal{L}_{\text{orth}}$ from (Yan et al., 2025b), extending it to ensure the active expert $e_a$ captures novel information distinct from established representations. Unlike (Yan et al., 2025b), which only constrains against the principal subspace $\mathbf{W}_M$, our $\mathcal{L}_{\text{orth}}$ enforces orthogonality against all previously trained semantic experts. Specifically, for the $i$-th expert $e_i$, we define the set of frozen indices as $\mathcal{I}_{\text{prev}} = \{M\} \cup \{0, \ldots, i-1\}$ and formulate the loss as:

$$\mathcal{L}_{\text{orth}} = \sum_{j \in \mathcal{I}_{\text{prev}}} \left( \|\mathbf{U}_i^T \mathbf{U}_j\|_F^2 + \|\mathbf{V}_i^T \mathbf{V}_j\|_F^2 \right), \qquad (5)$$

where $\mathbf{U}_i$ and $\mathbf{V}_i$ are the orthogonal bases for the active expert $e_i$, and $\{\mathbf{U}_j, \mathbf{V}_j\}_{j \in \mathcal{I}_{\text{prev}}}$ are the bases of the principal subspace and all previously trained experts. The **Appendix** provides a theoretical analysis of how the orthogonality constraint ensures feature diversity by limiting new updates within existing subspaces.

### 3.2.2. STAGE 2: ROUTER TRAINING AND SYSTEM INTEGRATION

After all $N_S$ semantic experts are specialized, their trained residual components are frozen. We then concurrently train the gating network $\mathcal{R}$ and the re-initialized classification head to integrate the full system. The optimization objective is threefold:

$$\mathcal{L}_{\text{Stage2}} = \mathcal{L}_{\text{cls}} + \lambda_2 \mathcal{L}_{\text{gating}} + \lambda_3 \mathcal{L}_{\text{balance}}. \qquad (6)$$

This objective incorporates three components. The primary classification loss ($\mathcal{L}_{\text{cls}}$) and the supervised gating loss ($\mathcal{L}_{\text{gating}}$) are both implemented as standard CE losses. $\mathcal{L}_{\text{cls}}$ is applied to the final real/fake prediction, while $\mathcal{L}_{\text{gating}}$ enforces routing correctness by using the ground-truth expert label $y_e$ for a given input $\mathbf{x}$. This supervised gating loss trains the router to output a sharp probability distribution centered on the target expert.

The load balancing loss, $\mathcal{L}_{\text{balance}}$, is an auxiliary regularizer adapted from (Fedus et al., 2022) to encourage router diversity:

$$\mathcal{L}_{\text{balance}} = N_S \sum_{i=1}^{N_S} \mathcal{F}_i \cdot \mathbf{P}_i. \qquad (7)$$

*Table 1.* Comparison of AIGI detection datasets, highlighting our proposed **Mirage** dataset. Legend: **Gen.Year** (newest generator year), **Num. (R/F)** (Real/Fake image count), **Wild** (in-the-wild), **Class.** (semantic classifications), **Min.Pairs** (semantically-close pairs), and **Real-Opt** (realism-optimized generators).

| Train-Dataset | Gen.Year | Num. (R/F) | Wild | Class. | Min.Pairs |
|---|---|---|---|---|---|
| CNNSpot (Wang et al., 2020) | ∼ 2018 | 360K/360K | × | ✓ | × |
| GenImage SDv1.4 (Zhu et al., 2023b) | ∼ 2022 | 162K/162K | × | × | × |
| GenImage (Zhu et al., 2023b) | ∼ 2022 | 1277K/1300K | × | × | × |
| DRCT-2M SDv1.4 (Chen et al., 2024) | ∼ 2023 | 118K/118K | × | × | ✓ |
| DRCT-2M (Chen et al., 2024) | ∼ 2023 | 118K/1892K | × | × | ✓ |
| **Mirage-Train** | ∼ 2025 | 933K/1674K | ✓ | ✓ | ✓ |

| Test-Dataset | Gen.Year | Num. (R/F) | Wild | Class. | Real-Opt |
|---|---|---|---|---|---|
| CNNSpot (Wang et al., 2020) | ∼ 2020 | 4K/4K | × | × | × |
| GenImage (Zhu et al., 2023b) | ∼ 2022 | 50K/50K | × | × | × |
| AIGCDetectBenchmark (Zhong et al., 2023) | ∼ 2023 | 76K/76K | × | × | × |
| DRCT-2M (Chen et al., 2024) | ∼ 2023 | 80K/80K | × | × | × |
| Chameleon (Yan et al., 2025a) | ∼ 2024 | 15K/11K | ✓ | × | × |
| **Mirage-Test** | ∼ 2025 | 22K/28K | ✓ | ✓ | ✓ |

For a batch $\mathcal{B}$ of size $|B|$, $\mathcal{F}_i = \frac{1}{|B|} \sum_{\mathbf{x}} \mathbb{I}(\arg\max(\mathbf{z}_{\mathbf{x}}) = i)$ is the fraction of inputs routed to expert $i$, and $\mathbf{P}_i = \frac{1}{|B|} \sum_{\mathbf{x}} \text{Softmax}(\mathbf{z}_{\mathbf{x}})_i$ is the average router probability allocated to expert $i$.

## 4. Mirage Dataset

The generalization of AIGI detectors is intrinsically linked to their training data. To address the limitations of existing benchmarks, we introduce **Mirage**, a novel, large-scale data foundation for training and validating detectors against contemporary generative threats. A comprehensive comparison with existing datasets is provided in Table 1.

### 4.1. Limitations of Existing Benchmarks

Current AIGI detection research is hindered by a reliance on outdated datasets with two primary limitations: **(1) Obsolete Generators and Content Gaps.** Most benchmarks comprise images from legacy models (e.g., GANs (Goodfellow et al., 2014), early DMs (Rombach et al., 2022)). Detectors trained on these may excel on established leaderboards but fail against modern "in-the-wild" threats, offering limited real-world utility. This is exacerbated by a lack of diversity; for instance, GenImage (Zhu et al., 2023b) omits crucial domains like anime and stylized art. **(2) Flawed Training Protocols.** Furthermore, many benchmarks mandate training on a single generator, a practice insufficient for capturing diverse forgery traces (Yan et al., 2025a).

### 4.2. Mirage-Train

To address these limitations, we introduce **Mirage-Train**, the large-scale, content-diverse training component of our **Mirage** data foundation. Its construction is guided by three principles: (1) **High Quality** (high-resolution, low-artifact images); (2) **Model Contemporaneity** (inclusion of recent generative models); and (3) **Ecological Validity** (data reflecting real-world scenarios).

### 4.2.1. SEMANTIC COMPOSITION AND DATA SOURCING

**Mirage-Train** aligns with the diverse categories identified in Chameleon (Yan et al., 2025a) (Human, Animal, Object, and Scene) while further incorporating Anime to address the prevalent distributions in synthetic imagery. We specifically introduce Anime to bridge the significant generalization gap observed between anime and photorealistic domains (Figures 2a and 2b), a critical distinction often overlooked in existing benchmarks.

**Real Image Collection.** We source high-resolution photographs from public collections (e.g., Pexels (Pexels)) to establish a photorealistic base. This is supplemented by a large corpus of human-created digital and anime art curated from online communities to comprehensively cover the stylized domain.

**Synthetic Image Collection.** We generate a vast image set using a diverse array of SOTA Text-to-Image (T2I) models, including prominent open-source generators (e.g., SD3.5 (Stability AI, 2024), Flux.1 (Black Forest Labs, 2024)) and commercial closed-source APIs. The fake image generation process uses a real-image-anchored prompting pipeline rather than randomly sampled or manually written free-form prompts: for each collected real image, we use an LMM annotator to produce a content description and a coarse semantic label (Human, Animal, Object, Scene, or Anime), and then use the content description as the prompt for generating the corresponding synthetic image with diverse T2I generators. This strategy preserves coarse semantic alignment between real and fake images within each category and ensures that prompts reflect naturally occurring visual content rather than artificial prompt templates. To ensure ecological validity, we further curate a large corpus of in-the-wild synthetic images from open web sources and third-party creation platforms.

**Purified Artifact Set.** Finally, to train our Universal Artifact Expert, we construct a purified artifact dataset featuring semantically identical real-fake pairs. Following (Rajan et al., 2024), we use *MS-COCO* (Lin et al., 2014) for real images and generate synthetic counterparts via reconstruction. We employ diverse VAEs, including those from SDv1.x–SD3.5 (Rombach et al., 2022) and specialized models such as 'TAESD' and 'TAESDXL' (Bohan, 2023), to encourage the expert to capture shared reconstruction artifacts rather than overfitting to a single VAE signature.

### 4.3. Mirage-Test

To rigorously evaluate robustness against in-the-wild threats, we introduce **Mirage-Test**. Unlike filtered web-based datasets like Chameleon (Yan et al., 2025a), Mirage-Test is a source-derived test set constructed from held-out SOTA generators. These models are optimized for maximum pho-

torealism via specialized fine-tunes, LoRA (Hu et al., 2022) modules, and proprietary data, establishing a more rigorous benchmark for high-fidelity, real-world threats.

## 5. Experiments

Comprehensive experiments confirm OmniAID's generalization. Robustness, computational efficiency, and further analyses are detailed in the **Appendix**.

### 5.1. Evaluation Setup

**Evaluation Protocol.** To ensure a fair comparison, we follow the protocol of (Zhu et al., 2023b; Yan et al., 2025a), training all models (including our standard OmniAID) exclusively on the GenImage-SD v1.4 dataset to assess generalization from a limited, standard benchmark. Alongside this, to evaluate performance in a realistic, modern scenario, we also train our OmniAID-Mirage model on our modern Mirage-Train dataset. All models are then evaluated on the GenImage (Zhu et al., 2023b) test set, the in-the-wild Chameleon (Yan et al., 2025a) dataset, and our new Mirage-Test. To further demonstrate the powerful detection performance of our OmniAID-Mirage, additional experiments on other benchmarks (Zhong et al., 2023; Chen et al., 2024) are provided in the **Appendix**.

**Evaluation Metrics.** We primarily report classification Accuracy (%), with additional results including Average Precision (AP), F1-score, and False Negative Rate (FNR) detailed in the **Appendix**.

**Implementation Details.** We use a pretrained **CLIP-ViT-L/14@336px** (Radford et al., 2021) backbone from OpenAI (OpenAI, 2021). Input images are resized to $512 \times 512$ to mitigate size variance before being adjusted to the required $336 \times 336$ resolution. Training uses AdamW (Loshchilov & Hutter, 2017) ($lr = 2 \times 10^{-4}$, batch size 32) for 1 epoch per stage on 4 NVIDIA H200 GPUs. For GenImage-SDv1.4, classes are grouped into two categories ('Human/Animal', 'Object/Scene') due to sparse classes, and use SDv1.4 VAE for the artifact set. Training takes 3 hours for GenImage and 18 hours for Mirage. Additional details are in the **Appendix**.

### 5.2. Benchmark Performance Evaluation

We compare OmniAID against a comprehensive set of SOTA AIGI detectors. These include (1) artifact-specific methods focused on low-level generator fingerprints (Wang et al., 2020; Zhang et al., 2019a; Frank et al., 2020; Ju et al., 2022; Liu et al., 2022; Zhong et al., 2023; Wang et al., 2023; Tan et al., 2024b), and (2) VFM-based generalizable methods that leverage large pre-trained models for robust detection (Ojha et al., 2023; Yan et al., 2025a;b).

*Table 2.* Performance (Accuracy %) on the **GenImage** benchmark. To ensure a fair comparison, all models (including OmniAID) are trained on GenImage-SD v1.4, except OmniAID-Mirage (on **Mirage-Train**). **Best** and underline{second-best} results are marked.

| Method | Midjourney | SD v1.4 | SD v1.5 | ADM | GLIDE | Wukong | VQDM | BigGAN | *Mean* |
|---|---|---|---|---|---|---|---|---|---|
| CNNSpot (Wang et al., 2020) | 52.8 | 96.3 | 95.9 | 50.1 | 39.8 | 78.6 | 53.4 | 46.8 | 64.2 |
| Spec (Zhang et al., 2019b) | 52.0 | 99.4 | 99.2 | 49.7 | 49.8 | 94.8 | 55.6 | 49.8 | 68.8 |
| F3Net (Qian et al., 2020) | 50.1 | 99.9 | **99.9** | 49.9 | 50.0 | **99.9** | 49.9 | 49.9 | 68.7 |
| GramNet (Zhang et al., 2019a) | 54.2 | 99.2 | 99.1 | 50.3 | 54.6 | 98.9 | 50.8 | 51.7 | 69.9 |
| DIRE (Wang et al., 2023) | 60.2 | 99.9 | 99.8 | 50.9 | 55.0 | 99.2 | 50.1 | 50.2 | 70.7 |
| UnivFD (Ojha et al., 2023) | 91.5 | 96.4 | 96.1 | 58.1 | 73.4 | 94.5 | 67.8 | 57.7 | 79.5 |
| GenDet (Zhu et al., 2023a) | 89.6 | 96.1 | 96.1 | 58.0 | 78.4 | 92.8 | 66.5 | 75.0 | 81.6 |
| PatchCraft (Zhong et al., 2023) | 79.0 | 89.5 | 89.3 | 77.3 | 78.4 | 89.3 | 83.7 | 72.4 | 82.3 |
| NPR (Tan et al., 2024b) | 81.0 | 98.2 | 97.9 | 76.9 | 89.8 | 96.9 | 84.1 | 84.2 | 88.6 |
| FatFormer (Liu et al., 2024) | 92.7 | **100.0** | **99.9** | 75.9 | 88.0 | **99.9** | **98.8** | 55.8 | 88.9 |
| DRCT (Chen et al., 2024) | 91.5 | 95.0 | 94.4 | 79.4 | 89.2 | 94.7 | 90.0 | 81.7 | 89.5 |
| AIDE (Yan et al., 2025a) | 79.4 | 99.7 | 99.8 | 78.5 | 91.8 | 98.7 | 80.3 | 66.9 | 86.9 |
| Effort (Yan et al., 2025b) | 82.4 | 99.8 | 99.8 | 78.7 | 93.3 | 97.4 | 91.7 | 77.6 | 91.1 |
| OmniAID | 85.7 | 98.9 | 98.8 | **91.4** | **98.7** | 98.1 | 97.3 | **98.7** | 95.9 |
| OmniAID-Mirage | **98.0** | 98.7 | 98.4 | 89.5 | 98.3 | 98.6 | 98.4 | 98.1 | 97.2 |

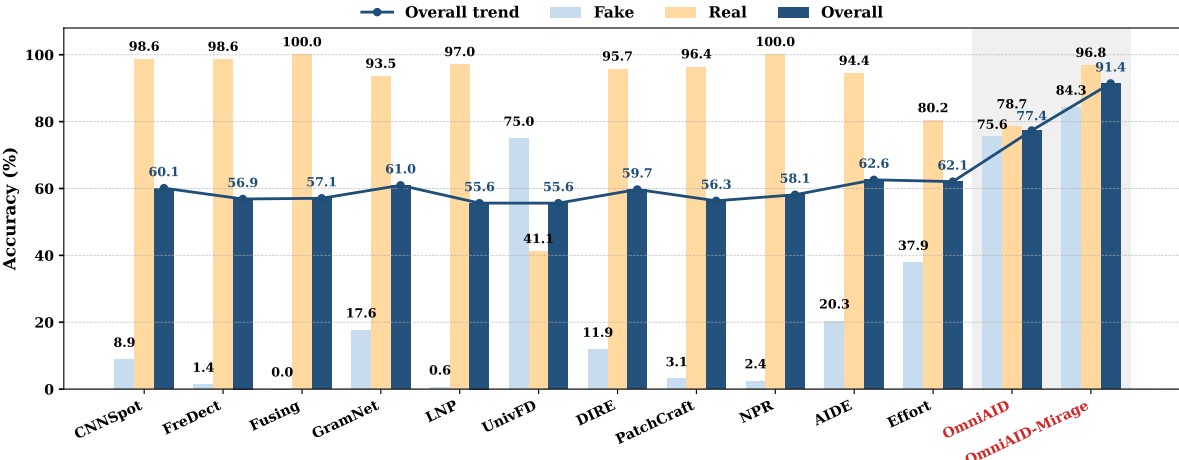

*Figure 5.* Performance (Accuracy %) comparison on the in-the-wild **Chameleon** benchmark. To ensure a fair comparison, all models (including OmniAID) are trained on GenImage-SD v1.4, except OmniAID-Mirage (on **Mirage-Train**).

### 5.2.1. COMPARISON ON GENIMAGE

On GenImage (Table 2), our standard OmniAID (trained on GenImage-SDv1.4) achieves 95.9% mean accuracy, significantly outperforming the SOTA Effort (91.1%). Our decoupled architecture shows superior generalization to unseen GANs (BigGAN: 98.7% vs. 77.6%) and diffusion models (ADM: 91.4% vs. 78.7%), even when trained on limited, outdated data. Furthermore, our OmniAID-Mirage achieves the highest accuracy (97.2%), demonstrating both SOTA performance and excellent backward compatibility.

### 5.2.2. COMPARISON ON CHAMELEON

On the in-the-wild Chameleon (Yan et al., 2025a) benchmark Figure 5, GenImage-trained detectors suffer a severe performance collapse, exhibiting a pronounced **Real/Fake detection bias**. Baselines like Fusing and NPR achieve high 'Real' (up to 100.0%) accuracy but catastrophic 'Fake' ac-

curacy (near 0.0%), indicating critical overfitting to specific GenImage artifacts. Unable to generalize, they misclassify novel Chameleon fakes as 'Real'. In stark contrast, our standard OmniAID achieves a balanced 77.4% mean accuracy (78.7% Real, 75.6% Fake). Critically, OmniAID-Mirage sets a new SOTA at 91.4%, demonstrating the robust, balanced detection essential for practical deployment.

### 5.2.3. COMPARISON ON MIRAGE-TEST

On our most challenging **Mirage-Test** Tables 3 and 4, featuring unseen high-fidelity generators, all GenImage-trained baselines fail dramatically (e.g., Effort, 43.03%). This confirms that existing benchmarks are inadequate for modern threats. Our standard OmniAID (trained on GenImage) performs better (51.10%) but is still fundamentally limited by its outdated training data. In contrast, OmniAID-Mirage achieves a superior 88.39% mean accuracy with consistent

*Table 3.* Performance (Accuracy %) on our **Mirage-Test**. To ensure a fair comparison, all models (including OmniAID) are trained on GenImage-SD v1.4, except OmniAID-Mirage (on **Mirage-Train**). *Note: Due to copyright considerations, the 'Anime' category consists solely of generated samples.*

| Method | Human | | | Animal | | | Object | | | Scene | | | Anime | | | *Mean* |
|---|---|---|---|---|---|---|---|---|---|---|---|---|---|---|---|---|
| | Real | Fake | Overall | Real | Fake | Overall | Real | Fake | Overall | Real | Fake | Overall | Real | Fake | Overall | |
| DIRE (Wang et al., 2023) | **99.07** | 1.22 | 50.14 | **99.20** | 2.60 | 50.90 | **97.18** | 1.33 | 49.26 | **98.80** | 0.58 | 49.69 | - | 2.18 | 2.18 | 40.43 |
| NPR (Tan et al., 2024b) | 79.32 | 12.17 | 45.74 | 68.86 | 17.91 | 43.39 | 77.12 | 12.67 | 44.89 | 71.92 | 18.00 | 44.96 | - | 13.45 | 13.45 | 38.49 |
| DRCT (Chen et al., 2024) | 90.20 | 6.12 | 48.16 | 93.43 | 13.77 | 53.60 | 91.58 | 7.17 | 49.38 | 91.97 | 5.63 | 48.80 | - | 10.28 | 10.28 | 42.04 |
| AIDE (Yan et al., 2025a) | 62.93 | 10.02 | 36.48 | 61.94 | 15.37 | 38.66 | 54.97 | 10.00 | 32.49 | 67.28 | 10.00 | 38.64 | - | 10.02 | 10.02 | 31.25 |
| Effort (Yan et al., 2025b) | 67.98 | 24.07 | 46.03 | 81.26 | 27.57 | 54.41 | 57.72 | 21.38 | 39.55 | 64.90 | 33.12 | 49.01 | - | 26.13 | 26.13 | 43.03 |
| OmniAID | 76.40 | 42.35 | 59.38 | 82.63 | 29.17 | 55.90 | 82.60 | 15.43 | 49.02 | 80.60 | 12.45 | 46.53 | - | 44.67 | 44.67 | 51.10 |
| OmniAID-Mirage | 98.13 | **89.25** | **93.69** | 93.69 | **69.06** | **81.37** | 97.17 | **72.67** | **84.92** | 98.53 | **75.60** | **87.07** | - | **94.92** | **94.92** | **88.39** |

*Table 4.* Performance (Average Precision %) on our **Mirage-Test**.

| Method | Human | Animal | Object | Scene | Anime | *Mean* |
|---|---|---|---|---|---|---|
| DIRE (Wang et al., 2023) | 47.00 | 54.69 | 42.48 | 42.75 | - | 46.73 |
| NPR (Tan et al., 2024b) | 45.54 | 44.16 | 44.69 | 45.17 | - | 44.89 |
| DRCT (Chen et al., 2024) | 42.44 | 55.13 | 43.41 | 44.18 | - | 46.29 |
| AIDE (Yan et al., 2025a) | 39.49 | 39.10 | 36.64 | 38.86 | - | 38.52 |
| Effort (Yan et al., 2025b) | 45.12 | 56.25 | 39.10 | 46.86 | - | 46.83 |
| OmniAID | 64.73 | 59.07 | 46.57 | 43.20 | - | 53.39 |
| OmniAID-Mirage | **99.18** | **92.57** | **97.02** | **98.47** | - | **96.81** |

*Table 5.* Ablation study on the core components of our hybrid MoE architecture. $e_0$ and $e_1$ are semantic experts for 'Human/Animal' and 'Object/Scene' (grouped due to class sparsity), while $e_U$ is the universal expert. All models are trained on GenImage-SD v1.4.

| Module | | | GenImage | Chameleon | Mirage-Test |
|---|---|---|---|---|---|
| $e_0$ | $e_1$ | $e_U$ | | | |
| ✓ | ✗ | ✗ | 84.38 | 58.86 | 39.63 |
| ✗ | ✓ | ✗ | 85.18 | 59.01 | 36.31 |
| ✗ | ✗ | ✓ | 83.31 | 60.85 | 45.14 |
| ✓ | ✓ | ✗ | 92.15 | 66.07 | 44.51 |
| ✓ | ✗ | ✓ | 91.90 | 68.11 | 47.35 |
| ✗ | ✓ | ✓ | 93.52 | 70.80 | 48.99 |
| ✓ | ✓ | ✓ | **95.94** | **77.35** | **51.10** |

performance across all categories. This proves the dual effectiveness of our specialized expert design and the absolute necessity of a modern, diverse training dataset.

## 5.3. Ablation Studies and Analysis

We conduct core component ablations on the OmniAID model trained on GenImage-SDv1.4, using this smaller benchmark to efficiently isolate our architectural contributions from the data-driven gains of our Mirage-Train dataset. In addition, to analyze dataset impact, we compare our OmniAID against previous SOTA methods, AIDE and Effort, trained on both GenImage and our Mirage-Train. Further ablations (e.g., on hyperparameters and loss functions) are available in the **Appendix**.

### 5.3.1. ANALYSIS OF HYBRID MoE DESIGN

We analyze our hybrid expert pool in Table 5.

**Key Insights: (1)** The full model (Row 7: $e_0 + e_1 + e_U$) achieves the best performance across all benchmarks, validating that the complete synergy of our dual decoupling (both *between* semantic domains and *between* semantics/artifacts) is crucial for maximum robustness. **(2)** The Universal Artifact Expert ($e_U$) is the most critical component for generalization. Removing it (Row 4) causes the largest OOD performance drop (11.28% on Chameleon), far exceeding the removal of any single semantic expert (Rows 5-6). This suggests semantic experts ($e_0, e_1$) are more prone to overfitting on domain-specific flaws, while $e_U$ captures more generalizable, low-level artifacts. **(3)** Comparing semantic experts, removing $e_1$ ('Object/Scene', Row 5) is more detrimental to OOD performance than removing $e_0$

('Human/Animal', Row 6). This finding is consistent with Figures 2a and 2b, where 'Object/Scene' domains showed better cross-domain generalization. We posit this is because models trained on strong, salient subjects (Human/Animal) are more susceptible to semantic overfitting, diminishing their contribution to generalization compared to the more diverse 'Object/Scene' expert.

### 5.3.2. ANALYSIS OF MIRAGE-TRAIN

Table 6 validates both our data and model contributions. First, it demonstrates the inadequacy of older data: training on our modern Mirage-Train (bottom block) universally and dramatically boosts in-the-wild detection performance (e.g., gains of +21.0% on Chameleon and +45.5% on Mirage for AIDE) compared to training on GenImage-SDv1.4 (top block). Second, it confirms our model's architectural superiority. While OmniAID-Mirage achieves the best overall performance across all benchmarks, competitors like Effort suffer from negative transfer (Effort-Mirage at 85.00% vs. Effort at 91.10% on GenImage). In contrast, our standard OmniAID shows far greater robustness, even outperforming AIDE-Mirage and Effort-Mirage on the GenImage, AIGCDetection and DRCT-2M.

### 5.3.3. FEATURE SPACE DECOUPLING VISUALIZATION

To empirically validate our decoupling hypothesis, we employ t-SNE to visualize the feature embeddings in Figure 6. **(a)** The Effort (Yan et al., 2025b) baseline exhibits a highly

*Table 6.* Performance (Accuracy %) comparing models trained on GenImage-SDv1.4 vs. our Mirage-Train.

| Method | GenImage | Chameleon | Mirage | AIGCDetection | DRCT-2M |
|---|---|---|---|---|---|
| AIDE | 86.88 | 62.60 | 31.25 | 82.20 | 64.22 |
| Effort | 91.10 | 62.06 | 43.03 | 86.36 | 62.96 |
| OmniAID | 95.94 | 77.35 | 51.10 | 88.87 | 88.21 |
| AIDE-Mirage | 92.46 | 83.61 | 76.78 | 86.73 | 79.76 |
| Effort-Mirage | 85.00 | 82.05 | 81.64 | 86.88 | 82.13 |
| OmniAID-Mirage | **97.24** | **91.42** | **88.39** | **92.88** | **91.91** |

entangled feature space, confirming that monolithic models learn a confused, mixed representation of Real/Fake samples and semantic categories. **(b)** In stark contrast, OmniAID exhibits a well-structured embedding space, demonstrating clear **Real vs. Fake Separation** within categories and tight, distinct **Semantic Clustering** (e.g., Human, Animal, Anime). This provides strong qualitative evidence that our hybrid MoE design successfully disentangles semantic representations from forgery artifacts.

### 5.3.4. ROUTER VISUALIZATION

We visualize the router's gating weights in Figure 7 to verify its internal mechanism. The router correctly dispatches inputs to their corresponding semantic experts: for example, a 'Human' image (center) assigns a 0.94 weight to the Human expert, while an 'Animal with Human' image (left) activates both the Animal (0.69) and Human (0.31) experts. This provides clear evidence that our two-stage training strategy successfully learns the intended expert specializations, rather than functioning as an uninterpretable black box.

To further probe unseen semantics, we additionally evaluate OmniAID on a small medical-image subset that is not included in the training taxonomy. The model achieves 92.0% accuracy on 400 samples (200 real and 200 fake). Its average routing distribution remains semantically plausible (Human: 0.37, Animal: 0.00, Object: 0.57, Scene: 0.06, Anime: 0.00), aligning with the presence of medical instruments, scans, and human anatomy. This result does not imply exhaustive open-set coverage, but it indicates that the router can reuse the most relevant existing experts while the universal artifact expert remains active.

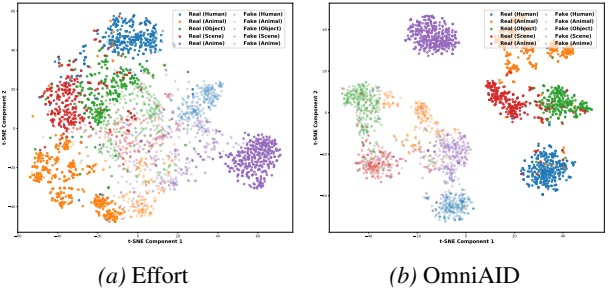

*(a)* Effort        *(b)* OmniAID

*Figure 6.* t-SNE visualization of feature decoupling on unseen test samples. Both models are trained on our Mirage-Train dataset.

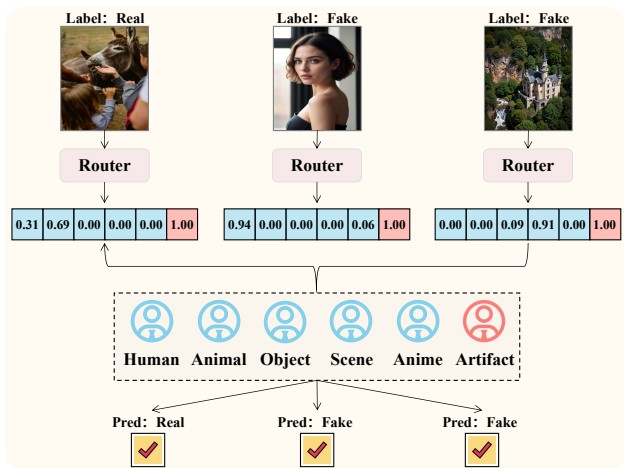

*Figure 7.* Visualization of the OmniAID routing mechanism.

## 6. Conclusion

In this work, we propose **OmniAID**, a novel MoE framework that fundamentally addresses the entanglement of semantic flaws and generator artifacts in universal AIGI detection. Our hybrid MoE architecture achieves robust decoupling by composing Routable Specialized Semantic Experts with a Fixed Universal Artifact Expert in an orthogonal subspace, optimized via a bespoke two-stage training strategy. Concurrently, we introduced **Mirage**, a modern dataset addressing the limitations of outdated benchmarks. Extensive experiments demonstrate that OmniAID establishes a new state-of-the-art, achieving superior generalization against modern, in-the-wild threats and validating the efficacy of our decoupling paradigm.

## Acknowledgements

This work was partially supported by Guangdong S&T Program (2024B0101040005), National Natural Science Foundation of China (Grant No. T2125006, 62571560) and Shanghai Artificial Intelligence Laboratory.

## Impact Statement

As AI-generated images become increasingly photorealistic, the potential for misuse in misinformation, fraud, and identity manipulation grows substantially. This paper contributes a more robust detection framework to help safeguard digital integrity. Our decoupled design reduces semantic biases that monolithic detectors often exhibit against specific demographics or styles, promoting fairer deployment. We emphasize that our work targets the detection of deceptive misuse rather than restricting legitimate creative applications, and recommend integrating such tools within human-in-the-loop workflows for responsible decision-making.

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

## A. Information-Theoretic Motivation for Decoupling

This section provides a heuristic information-theoretic motivation rather than a strictly rigorous mathematical proof for our decoupled design. We view AIGI detection as extracting forgery traces under strong semantic nuisance variation, and use mutual information only as a proxy for comparing different data regimes.

### A.1. Problem Formulation and Variable Definition

For a generated image ($Y = 1$), we write

$$X = \mathcal{G}(S_{content}, F_{trace}) + \epsilon, \tag{8}$$

where $S_{content}$ denotes high-level semantic content, $F_{trace}$ denotes intrinsic traces introduced by the generative process, and $\epsilon$ denotes nuisance perturbations such as compression, resizing, and transmission noise. We decompose $F_{trace}$ into content-agnostic artifacts $A_{univ}$ and content-dependent anomalies $A_{spec}$. For real images ($Y = 0$), we use $F_{trace} = \emptyset$ as a notational convention. The detector should rely on cues associated with $F_{trace}$ while being robust to changes in $S_{content}$ and $\epsilon$.

### A.2. Motivation for Semantic Experts: SNR Enhancement

The core difficulty in generalizable detection is that standard models often overfit to $S_{content}$ rather than capturing $F_{trace}$. We quantify this difficulty using the Signal-to-Noise Ratio (SNR) of information.

**Definition A.1** (Information SNR). The capability of a model to learn robust features on a dataset $\mathcal{D}$ is characterized by the ratio of mutual information provided by the forgery traces versus the semantic content:

$$\text{SNR}(\mathcal{D}) \triangleq \frac{I_{\mathcal{D}}(X; F_{trace})}{I_{\mathcal{D}}(X; S_{content})}. \tag{9}$$

A higher SNR implies that the informative bits related to forgery are more dominant relative to the bits related to semantics, facilitating robust learning. To simplify this metric, we first approximate mutual information using entropy.

**Approximation Logic.** We decompose the mutual information as $I(X; Z) = H(Z) - H(Z|X)$. In the context of high-fidelity AIGI detection, we observe that the image $X$ is a primary manifestation of the latent factors $S$ and $F$. Under the condition where the generative process $\mathcal{G}$ preserves the essential characteristics of these latent variables, the conditional uncertainty $H(Z|X)$ becomes negligible relative to the marginal entropy $H(Z)$. Consequently, the mutual information is dominated by the marginal entropy of the latent factors, allowing us to adopt the following analytical simplifications:

$$\begin{aligned} I_{\mathcal{D}}(X; S_{content}) &\approx H_{\mathcal{D}}(S_{content}), \\ I_{\mathcal{D}}(X; F_{trace}) &\approx H_{\mathcal{D}}(F_{trace}). \end{aligned} \tag{10}$$

By substituting these into Eq. (9), the Information SNR can be effectively analyzed through the ratio of entropies:

$$\text{SNR}(\mathcal{D}) \approx \frac{H_{\mathcal{D}}(F_{trace})}{H_{\mathcal{D}}(S_{content})}. \tag{11}$$

*Remark* A.2 (Justification for Relative Analysis). This approximation is only used to compare the relative signal regimes before and after semantic partitioning. Although perturbations such as compression may increase the absolute conditional uncertainty $H(Z|X)$, they affect both the global dataset and semantic subsets. The key quantity is therefore the relative reduction of semantic variation versus trace variation.

Based on this, we analyze the Semantic Router $\mathcal{R}$, which partitions the global dataset $\mathcal{D}_{global}$ into semantic subsets $\mathcal{D}_k$.

**Assumption A.3** (Differential Entropy Reduction). Let $\rho_S$ and $\rho_F$ denote the entropy retention ratios for semantics and traces in a subset $\mathcal{D}_k$ relative to the global dataset:

$$\rho_S = \frac{H_{\mathcal{D}_k}(S_{content})}{H_{\mathcal{D}_{global}}(S_{content})}, \quad \rho_F = \frac{H_{\mathcal{D}_k}(F_{trace})}{H_{\mathcal{D}_{global}}(F_{trace})}. \tag{12}$$

Since the router explicitly clusters data based on semantic criteria, it enforces *inter-class semantic separation*. While intra-class diversity remains, the explicit removal of other semantic categories results in a substantial reduction of semantic entropy, i.e., $\rho_S \ll 1$. In contrast, forgery traces $F_{trace}$ consist of both content-agnostic universal artifacts ($A_{univ}$) and content-dependent flaws ($A_{spec}$). While semantic filtering may reduce the diversity of $A_{spec}$ from other domains, the universal artifacts $A_{univ}$ are largely preserved. Consequently, the suppression of semantic diversity is significantly more dominant than that of trace diversity, implying the inequality $\rho_S \ll \rho_F$.

**Proposition A.4** (Local SNR Enhancement). *Under Assumption A.3, training a specialized expert on a semantic subset $\mathcal{D}_k$ yields an enhanced Signal-to-Noise Ratio compared to training on the global dataset $\mathcal{D}_{global}$:*

$$\mathrm{SNR}(\mathcal{D}_k) \gg \mathrm{SNR}(\mathcal{D}_{global}). \tag{13}$$

*Analysis.* We compare the SNR of the expert subset against the global SNR by substituting the scaling factors from Assumption A.3:

$$\begin{aligned}
\mathrm{SNR}(\mathcal{D}_k) &\approx \frac{H_{\mathcal{D}_k}(F_{trace})}{H_{\mathcal{D}_k}(S_{content})} \\
&= \frac{\rho_F \cdot H_{\mathcal{D}_{global}}(F_{trace})}{\rho_S \cdot H_{\mathcal{D}_{global}}(S_{content})} \\
&= \left(\frac{\rho_F}{\rho_S}\right) \cdot \mathrm{SNR}(\mathcal{D}_{global}).
\end{aligned} \tag{14}$$

Under Assumption A.3, the semantic clustering ensures that the semantic entropy decays significantly faster than the artifact entropy ($\rho_S \ll \rho_F$), implying $\frac{\rho_F}{\rho_S} \gg 1$. Consequently, $\mathrm{SNR}(\mathcal{D}_k) \gg \mathrm{SNR}(\mathcal{D}_{global})$. This suggests a more favorable signal regime for capturing intrinsic forgery traces within each expert. $\square$

*Remark* A.5 (Impact of Semantic Filtering on Artifacts). Filtering for a specific semantic domain can remove content-dependent artifacts from other domains, so $\rho_F$ is not assumed to be one. The local SNR argument only requires a relative effect: because the router clusters by semantics rather than artifacts, semantic diversity is expected to decrease more strongly than trace diversity.

## A.3. Motivation for Universal Expert: Conditional Independence

While semantic experts maximize SNR, we also need to capture purely content-agnostic artifacts ($A_{univ}$). To achieve this without semantic interference, we employ the *Purified Artifact Set $\mathcal{D}_{pair}$*, consisting of real-reconstruction pairs ($x_{real}, x_{rec}$). We argue that this paired training strategy discourages reliance on semantic content through approximate conditional independence.

**Proposition A.6** (Approximate Semantic-Label Independence). *On $\mathcal{D}_{pair}$, if reconstruction preserves high-level semantics, the label $Y$ is approximately independent of semantic content $S_{content}$.*

*Analysis.* We demonstrate this by analyzing the mutual information $I(Y; S_{content}) = H(Y) - H(Y \mid S_{content})$. In a balanced binary classification task, the marginal entropy is maximal: $H(Y) = 1$ bit. For a real image with semantic content $s$, a high-fidelity reconstruction (e.g., using FLUX VAE (Black Forest Labs, 2024)) preserves the same high-level content up to small drift, so both labels occur with nearly equal probability conditioned on $s$:

$$P(Y = 1 \mid S_{content} = s) \approx P(Y = 0 \mid S_{content} = s) \approx 0.5. \tag{15}$$

Thus $H(Y \mid S_{content}) \approx 1$ and

$$I_{\mathcal{D}_{pair}}(Y; S_{content}) = H(Y) - H(Y \mid S_{content}) \approx 0. \tag{16}$$

Semantic content therefore provides negligible information for classification on the paired set. To minimize the classification loss, the discriminator is encouraged to focus on the residual discrepancy $\Delta(x_{rec}, x_{real})$, which corresponds to the universal artifacts $A_{univ}$. $\square$

*Remark* A.7 (Robustness to Semantic Drift). VAE reconstruction may introduce slight semantic drift, so the independence is approximate rather than exact. However, modern high-fidelity VAEs preserve high-level semantics well enough that the residual semantic-label correlation is expected to be much weaker than the artifact-label correlation introduced by reconstruction traces.

## B. Theoretical Analysis of Gradient Orthogonality and Feature Decoupling

In this section, we provide a theoretical analysis of how the proposed orthogonality constraints encourage feature diversity and prevent redundancy. While we utilize Singular Value Decomposition (SVD) to initialize the expert weights $W_R$ within subspaces orthogonal to the principal components, we adopt a relaxed optimization strategy during training. Specifically, the factors $U_R$ and $V_R$ are updated via standard gradient descent without enforcing strict self-orthogonality constraints (e.g., via Stiefel manifold optimization). This design choice is intended to maintain training stability and efficiency, effectively allowing the model to leverage SVD-based priors while permitting flexible low-rank adaptation. Consequently, the orthogonality loss $\mathcal{L}_{orth}$ functions as a soft regularizer for inter-expert diversity. Crucially, since previous experts are frozen during sequential training, the goal of minimizing this orthogonality error is not to prevent catastrophic forgetting, but to limit the projection of new updates onto existing subspaces. This encourages new experts to learn representations that are distinct from both the principal subspace and previously learned experts.

**Definition B.1** (SVD-Initialized Parameterization). Let the $i$-th residual expert be parameterized as $W_{R,i} = U_i \Sigma_i V_i^T$, where $U_i \in \mathbb{R}^{d_{out} \times r}$ and $V_i \in \mathbb{R}^{d_{in} \times r}$ are factor matrices, and $\Sigma_i \in \mathbb{R}^{r \times r}$ is a diagonal matrix. Crucially, these parameters are initialized via the SVD of the residual components of pre-trained weights, such that at training step $t = 0$:

$$U_i^{(0)T} U_i^{(0)} = I_r \quad \text{and} \quad V_i^{(0)T} V_i^{(0)} = I_r. \tag{17}$$

During optimization ($t > 0$), we adopt a *relaxed constraint* setting where strict self-orthogonality is not explicitly enforced, while the low-rank factorized structure is preserved via the decomposition architecture.

*Remark* B.2 (The Role of SVD Initialization). The SVD initialization serves as a strong subspace prior for our decoupling strategy. By initializing $W_{R,i}$ from the residual components of SVD, the expert starts outside the principal SVD subspace of $W_M$, ensuring zero initial Frobenius overlap with the principal component and reducing redundant reuse of the dominant base-model directions at the beginning of training. Although strict orthogonality is relaxed during gradient updates, this initialization provides a stable starting point for the subsequent soft orthogonality regularization. Empirically, Table 13 shows that SVD initialization yields slightly stronger OOD performance than standard LoRA initialization, consistent with its lower initial redundancy with the principal subspace.

**Definition B.3** (Redundant Update Projection). Let $\mathcal{L}_i$ denote the loss function for the active expert $i$, with the update direction $\Delta W_{R,i} = -\eta \nabla_{W_{R,i}} \mathcal{L}_i$. The *redundant projection* of this update onto the subspace of a frozen expert $j$ ($j < i$) is defined as:

$$\mathcal{P}(i \to j) \triangleq \left| \langle \Delta W_{R,i}, W_{R,j} \rangle_F \right| = \left| \text{Tr}(\Delta W_{R,i}^T W_{R,j}) \right|. \tag{18}$$

*Remark* B.4 (Interpretation of Projection Minimization). Unlike traditional continual learning where "interference" implies altering previous knowledge, here expert $j$ is frozen. Thus, minimizing $\mathcal{P}(i \to j)$ does not protect expert $j$, but rather constrains expert $i$. A large $\mathcal{P}(i \to j)$ implies that the gradient for expert $i$ is parallel to the existing weight $W_{R,j}$, meaning expert $i$ is attempting to re-learn features already captured by $j$. By bounding this term, we mitigate feature redundancy and enforce semantic diversity.

**Definition B.5** ($\epsilon$-Approximate Cross-Orthogonality). Experts $i$ and $j$ satisfy $\epsilon$-approximate cross-orthogonality if their subspaces remain nearly orthogonal:

$$\|U_i^T U_j\|_F \le \epsilon_U \quad \text{and} \quad \|V_i^T V_j\|_F \le \epsilon_V, \tag{19}$$

where $\epsilon_U, \epsilon_V \ge 0$ are small constants minimized by our loss function $\mathcal{L}_{orth}$.

**Assumption B.6** (Boundedness of Parameters and Updates). We assume the factor matrices and their updates remain bounded throughout optimization. Specifically, there exist finite constants $C_U, C_V, S_\Sigma, M_\Delta > 0$ such that for any expert $i$:

$$\|U_i\|_2 \le C_U, \quad \|V_i\|_2 \le C_V, \quad \|\Sigma_i\|_F \le S_\Sigma, \quad \text{and} \quad \|\Delta \cdot\|_F \le M_\Delta, \tag{20}$$

where $\|\Delta \cdot\|_F$ represents the update magnitude for any factor matrix ($U_i, \Sigma_i, V_i$).

*Remark* B.7 (Justification of Boundedness). These bounds are structurally enforced by our training protocol: 1) **Initialization**: SVD initialization ensures $\|U_i\|_2 = \|V_i\|_2 = 1$ at $t = 0$; 2) **Regularization**: Weight decay ($L_2$ penalty) actively counteracts scale ambiguity, preventing unbounded parameter growth; 3) **Stability**: Gradient clipping and a finite learning rate strictly constrain the step size $M_\Delta$, validating the first-order perturbation analysis.

**Proposition B.8** (First-Order Projection Bound under Relaxed Orthogonality). *Under Assumption B.6 and $\epsilon$-approximate cross-orthogonality, the redundant update projection satisfies*

$$\mathcal{P}(i \to j) \le C_1 \epsilon_V + C_2 \epsilon_U + C_3 \epsilon_U \epsilon_V + O(M_\Delta^2), \tag{21}$$

*where $C_1 = S_\Sigma^2 M_\Delta C_U$, $C_2 = S_\Sigma^2 M_\Delta C_V$, and $C_3 = M_\Delta S_\Sigma$ are constants induced by the bounds in Assumption B.6.*

*Proof.* Expanding the factorized weight update and retaining the first-order terms gives

$$\Delta W_{R,i} = (\Delta U_i)\Sigma_i V_i^T + U_i(\Delta\Sigma_i)V_i^T + U_i\Sigma_i(\Delta V_i)^T + O(M_\Delta^2). \tag{22}$$

For clarity, denote the three first-order components by

$$\Delta W_U = (\Delta U_i)\Sigma_i V_i^T, \quad \Delta W_\Sigma = U_i(\Delta\Sigma_i)V_i^T, \quad \Delta W_V = U_i\Sigma_i(\Delta V_i)^T. \tag{23}$$

Since the frozen expert is $W_{R,j} = U_j\Sigma_j V_j^T$, the redundant projection satisfies

$$\mathcal{P}(i \to j) = \left| \langle \Delta W_U + \Delta W_\Sigma + \Delta W_V + O(M_\Delta^2), W_{R,j} \rangle_F \right| \tag{24}$$
$$\le T_U + T_\Sigma + T_V + O(M_\Delta^2), \tag{25}$$

where

$$T_U = |\langle \Delta W_U, W_{R,j} \rangle_F|, \quad T_\Sigma = |\langle \Delta W_\Sigma, W_{R,j} \rangle_F|, \quad T_V = |\langle \Delta W_V, W_{R,j} \rangle_F|. \tag{26}$$

For the $U_i$ update term,

$$T_U = \left| \text{Tr}\left( V_i \Sigma_i^T (\Delta U_i)^T U_j \Sigma_j V_j^T \right) \right| \tag{27}$$
$$= \left| \text{Tr}\left( (V_j^T V_i)(\Sigma_i^T (\Delta U_i)^T U_j \Sigma_j) \right) \right| \tag{28}$$
$$\le \|V_j^T V_i\|_F \|\Sigma_i^T (\Delta U_i)^T U_j \Sigma_j\|_F \tag{29}$$
$$\le \epsilon_V \|\Sigma_i\|_2 \|\Delta U_i\|_F \|U_j\|_2 \|\Sigma_j\|_2 \tag{30}$$
$$\le S_\Sigma^2 M_\Delta C_U \epsilon_V. \tag{31}$$

For the $\Sigma_i$ update term,

$$T_\Sigma = \left| \text{Tr}\left( V_i (\Delta\Sigma_i)^T U_i^T U_j \Sigma_j V_j^T \right) \right| \tag{32}$$
$$= \left| \text{Tr}\left( (V_j^T V_i)((\Delta\Sigma_i)^T U_i^T U_j \Sigma_j) \right) \right| \tag{33}$$
$$\le \|V_j^T V_i\|_F \|(\Delta\Sigma_i)^T U_i^T U_j \Sigma_j\|_F \tag{34}$$
$$\le \epsilon_V \|\Delta\Sigma_i\|_F \|U_i^T U_j\|_F \|\Sigma_j\|_2 \tag{35}$$
$$\le M_\Delta S_\Sigma \epsilon_U \epsilon_V. \tag{36}$$

For the $V_i$ update term,

$$T_V = \left| \text{Tr}\left( (\Delta V_i)\Sigma_i^T U_i^T U_j \Sigma_j V_j^T \right) \right| \tag{37}$$
$$= \left| \text{Tr}\left( (U_i^T U_j)(\Sigma_j V_j^T (\Delta V_i)\Sigma_i^T) \right) \right| \tag{38}$$
$$\le \|U_i^T U_j\|_F \|\Sigma_j V_j^T (\Delta V_i)\Sigma_i^T\|_F \tag{39}$$
$$\le \epsilon_U \|\Sigma_j\|_2 \|V_j\|_2 \|\Delta V_i\|_F \|\Sigma_i\|_2 \tag{40}$$
$$\le S_\Sigma^2 M_\Delta C_V \epsilon_U. \tag{41}$$

Combining the three terms gives

$$\mathcal{P}(i \to j) \le C_1 \epsilon_V + C_2 \epsilon_U + C_3 \epsilon_U \epsilon_V + O(M_\Delta^2). \tag{42}$$

Thus, up to higher-order update terms, the redundant projection of a new expert update onto a frozen expert is controlled by the cross-orthogonality errors $\epsilon_U$ and $\epsilon_V$. Minimizing $\mathcal{L}_{orth}$ therefore reduces the tendency of the active expert to update along subspaces already occupied by previous experts, mitigating feature redundancy under the relaxed optimization scheme. □

---

**Algorithm 1** OmniAID Training Framework

---

1: **Input:** Datasets $\mathcal{D} = \{\mathcal{D}_1, \ldots, \mathcal{D}_N\}$ (including Semantic and Artifact subsets), Mixed Dataset $\mathcal{D}_{mix}$.
2: **Parameters:** Frozen Principal Model $\{W_M^{(l)}\}_{l=1}^L$ (with factors $U_M, \Sigma_M, V_M$), Trainable Experts $\{W_{R,i}^{(l)}\}_{l=1}^L$ for domains $i = 1 \ldots N$.
3: **Hyperparameters:** Rank $r$, Top-$k$, Weights $\lambda_1, \lambda_2, \lambda_3$.
4: 

---

5: **Stage 1: Expert Specialization via Hard-Sampling**
6: **for** domain $i = 1$ to $N$ **do**
7:    Re-initialize Classification Head. // Head is specific to current expert training
8:    **for** epoch $e = 1$ to $E_1$ **do**
9:       **for** mini-batch $(x, y)$ sampled from $\mathcal{D}_i$ **do**
10:         // 1. Forward Pass with Current Expert $i$
11:         // Calculate weights for all $L$ layers
12:         $W_{R,i}^{(l)} \leftarrow U_i^{(l)} \Sigma_i^{(l)} V_i^{(l)T}, \quad \forall l \in \{1, \ldots, L\}$
13:         $W_F^{(l)} \leftarrow W_M^{(l)} + W_{R,i}^{(l)}$
14:         $\hat{y} \leftarrow \text{Model}(x; \{W_F^{(l)}\}, \text{Head})$
15:         // 2. Compute Losses
16:         $\mathcal{L}_{cls} \leftarrow \text{CrossEntropy}(\hat{y}, y)$
17:         // Orthogonality: against Principal Space ($M$) AND Previous Experts ($j < i$)
18:         $\mathcal{S}_{prev} \leftarrow \{(U_M, V_M)\} \cup \{(U_j, V_j)\}_{j=1}^{i-1}$
19:         $\mathcal{L}_{orth} \leftarrow \sum_{(U_{pre}, V_{pre}) \in \mathcal{S}_{prev}} \sum_{l=1}^L \left( \|U_i^{(l)T} U_{pre}^{(l)}\|_F^2 + \|V_i^{(l)T} V_{pre}^{(l)}\|_F^2 \right)$
20:         Update $\{U_i, \Sigma_i, V_i\}$ and Head to minimize $\mathcal{L}_{cls} + \lambda_1 \mathcal{L}_{orth}$.
21:       **end for**
22:    **end for**
23: **end for**
24: 

---

25: **Stage 2: Router Training & System Integration**
26: Freeze all Experts. Identify Semantic Experts and Universal Expert.
27: Re-initialize Classification Head. // Reset Head for the aggregated system
28: **for** epoch $e = 1$ to $E_2$ **do**
29:    **for** mini-batch $(x, y, s)$ sampled from $\mathcal{D}_{mix}$ **do**
30:       // 1. Router Forward & Selection (for Semantic Experts)
31:       $z \leftarrow \mathcal{R}_\phi(x), \quad p(x) \leftarrow \text{Softmax}(z)$
32:       Select Top-$k$ semantic indices $\mathcal{T}$ and normalize weights $g(x)$.
33:       // 2. Weight Aggregation (Base + Universal + Routed Semantic)
34:       $W_{MoE}^{(l)} \leftarrow \sum_{t \in \mathcal{T}} g_t(x) \cdot W_{R,t}^{(l)}, \quad \forall l \in \{1, \ldots, L\}$
35:       $W_F^{(l)} \leftarrow W_M^{(l)} + W_{R,U}^{(l)} + W_{MoE}^{(l)}$
36:       $\hat{y} \leftarrow \text{Model}(x; \{W_F^{(l)}\}, \text{Head})$
37:       // 3. Compute Losses
38:       $\mathcal{L}_{cls} \leftarrow \text{CrossEntropy}(\hat{y}, y)$
39:       $\mathcal{L}_{gate} \leftarrow \text{CrossEntropy}(p(x), s)$
40:       Calculate importance $f_i = \frac{1}{|\mathcal{B}|} \sum \mathbb{I}(\text{argmax}(p(x)) = i)$ and mean $P_i = \frac{1}{|\mathcal{B}|} \sum p_i(x)$.
41:       $\mathcal{L}_{balance} \leftarrow N_{sem} \sum_i f_i \cdot P_i$
42:       Update Router $\phi$ and Head to minimize $\mathcal{L}_{cls} + \lambda_2 \mathcal{L}_{gate} + \lambda_3 \mathcal{L}_{balance}$.
43:    **end for**
44: **end for**

---

## C. Training Algorithm

We present the training algorithm for OmniAID as shown in Algorithm 1.

## D. More Implementation Details

We provide detailed hyperparameter settings for reproducibility. All models are trained and evaluated using 4 NVIDIA H200 GPUs.

**Implementation Details for Semantic Generalization of UnivFD (Ojha et al., 2023) and Effort (Yan et al., 2025b) (Figures 2a and 2b).** For this evaluation, we curate a specific subset from the **Mirage-Train** dataset, comprising 10,000 training images and 1,000 test images per semantic category. To mitigate potential biases introduced by specific generation architectures, both the training and test sets are synthesized using an ensemble of diverse generators. All hyperparameters for training UnivFD and Effort strictly adhere to the official open-source implementations to ensure a fair comparison.

**Configuration for GenImage-SDv1.4 (Zhu et al., 2023b).** For the OmniAID trained on the GenImage-SDv1.4 subset, we set the learning rates for Stage 1 (Expert Specialization) and Stage 2 (Router Training) to $2 \times 10^{-4}$ and $2 \times 10^{-5}$, respectively. Regarding data augmentation, we apply only standard resizing and normalization. The trainable rank $r$ for the SVD-based experts is fixed at 4. During both Stage 2 training and inference, the router activates the Top-1 ($K = 1$) semantic expert. The loss weighting hyperparameters are configured as $\lambda_1 = 0.01$, $\lambda_2 = 0.1$, and $\lambda_3 = 0.1$.

**Configuration for Mirage-Train.** Conversely, for the OmniAID trained on our large-scale Mirage-Train dataset, the learning rate is maintained at $2 \times 10^{-4}$ across both training stages. Consistent with the GenImage configuration, no additional data augmentation strategies are employed. To accommodate the higher diversity and complexity of the Mirage dataset, we increase the trainable rank $r$ to 8 and set the number of active experts to Top-2 ($K = 2$). Accordingly, the loss weights are adjusted to $\lambda_1 = 0.001$, $\lambda_2 = 0.1$, and $\lambda_3 = 0.001$.

## E. Experiments on More Benchmarks

To further validate the robust detection capabilities of OmniAID-Mirage, we extend our evaluation to two additional widely recognized benchmarks: AIGCDetectBenchmark (Zhong et al., 2023) and DRCT-2M (Chen et al., 2024).

### E.1. Comparison on AIGCDetectBenchmark

The AIGCDetectBenchmark predominantly comprises legacy GAN-based methods (e.g., ProGAN, StyleGAN) and early diffusion models. Evaluating OmniAID-Mirage, which is trained on modern generators, on this benchmark serves as a rigorous test of backward compatibility. As shown in Tables 7 and 8, despite the significant distributional shift between the training data and these legacy generators, OmniAID-Mirage achieves superior performance. Specifically, it surpasses Effort (Yan et al., 2025b) by a substantial margin in terms of mean accuracy. This result confirms that our disentangled representation avoids catastrophic forgetting of low-level artifact signatures while acquiring high-level semantic sensitivity, effectively bridging the gap between legacy and modern AI-generated image detection.

*Table 7.* Performance comparison on the AIGCDetectBenchmark. We report detection Accuracy (ACC %). All baselines are trained on ProGAN, whereas our OmniAID-Mirage is trained on the Mirage-Train. The **best** and second-best results are marked in bold and underline, respectively.

| Method | ProGAN | StyleGAN | BigGAN | CycleGAN | StarGAN | GauGAN | StyleGAN2 | WFIR | ADM | Glide | Midjourney | SDv1.4 | SDv1.5 | VQDM | Wukong | DALLE2 | SDXL | Mean |
|---|---|---|---|---|---|---|---|---|---|---|---|---|---|---|---|---|---|---|
| CNNSpot | **100.00** | 90.17 | 71.17 | 87.62 | 94.60 | 81.42 | 86.91 | 91.65 | 60.39 | 58.07 | 51.39 | 50.57 | 50.53 | 56.46 | 51.03 | 50.45 | 53.03 | 69.73 |
| FreDect | 99.36 | 78.02 | 81.97 | 78.77 | 94.62 | 80.57 | 66.19 | 50.75 | 63.42 | 54.13 | 45.87 | 38.79 | 39.21 | 77.80 | 40.30 | 34.70 | 51.23 | 63.28 |
| Fusing | **100.00** | 85.20 | 77.40 | 87.00 | 97.00 | 77.00 | 83.30 | 66.80 | 49.00 | 57.20 | 52.20 | 51.00 | 51.40 | 55.10 | 51.70 | 52.80 | 55.60 | 67.63 |
| LNP | 99.67 | 91.75 | 77.75 | 84.10 | 99.92 | 75.39 | 94.64 | 70.85 | 84.73 | 80.52 | 65.55 | 85.55 | 85.67 | 74.46 | 82.06 | 88.75 | 87.75 | 85.28 |
| LGrad | 99.83 | 91.08 | 85.62 | 86.94 | 99.27 | 78.46 | 85.32 | 55.70 | 67.15 | 66.11 | 65.35 | 63.02 | 63.67 | 72.99 | 59.55 | 65.45 | 71.30 | 75.11 |
| UnivFD | 99.81 | 84.93 | 95.08 | 98.33 | 95.75 | 99.47 | 74.96 | 86.90 | 66.87 | 62.46 | 56.13 | 63.66 | 63.49 | 85.31 | 70.93 | 50.75 | 50.73 | 76.80 |
| DIRE-G | 95.19 | 83.03 | 70.12 | 74.19 | 95.47 | 67.79 | 75.31 | 58.05 | 75.78 | 71.75 | 58.01 | 49.74 | 49.83 | 53.68 | 54.46 | 66.48 | 55.35 | 67.90 |
| DIRE-D | 52.75 | 51.31 | 49.70 | 49.58 | 46.72 | 51.23 | 51.72 | 53.30 | **98.25** | 92.42 | 89.45 | 91.24 | 91.63 | 91.90 | 90.90 | 92.45 | 91.28 | 72.70 |
| PatchCraft | **100.00** | 92.77 | 95.80 | 70.17 | 99.97 | 71.58 | 89.55 | 85.80 | 82.17 | 83.79 | 90.12 | 95.38 | 95.30 | 88.91 | 91.07 | 96.60 | 98.43 | 89.85 |
| NPR | 99.79 | 97.70 | 84.35 | 96.10 | 99.35 | 82.50 | **98.38** | 65.80 | 69.69 | 78.36 | 77.85 | 78.63 | 78.89 | 78.13 | 76.11 | 64.90 | 98.40 | 83.82 |
| AIDE | 99.99 | **99.64** | 83.95 | 98.48 | 99.91 | 73.25 | **98.00** | 94.20 | 93.43 | 95.09 | 77.20 | 93.00 | 92.85 | 95.16 | 93.55 | 96.60 | 97.05 | **93.03** |
| Effort | **100.00** | 95.80 | **99.58** | **99.66** | **99.98** | **99.84** | 92.55 | **94.60** | 70.68 | 64.61 | 50.03 | 55.23 | 55.21 | 76.55 | 56.77 | 53.05 | 50.13 | 77.31 |
| OmniAID-Mirage | 80.90 | 90.77 | 82.80 | 90.28 | 98.90 | 83.58 | 86.81 | 88.05 | 89.50 | **98.34** | **98.01** | **98.69** | **98.36** | **98.38** | **98.60** | **98.20** | **98.85** | 92.88 |

*Table 8.* Performance comparison on the AIGCDetectBenchmark. We report detection Average Precision (AP %). All baselines are trained on ProGAN, whereas our OmniAID-Mirage is trained on the Mirage-Train. The **best** and second-best results are marked in bold and underline, respectively.

| Method | ProGAN | StyleGAN | BigGAN | CycleGAN | StarGAN | GauGAN | StyleGAN2 | WFIR | ADM | Glide | Midjourney | SDv1.4 | SDv1.5 | VQDM | Wukong | DALLE2 | SDXL | Mean |
|---|---|---|---|---|---|---|---|---|---|---|---|---|---|---|---|---|---|---|
| CNNSpot | **100.00** | 99.83 | 85.99 | 94.94 | 99.04 | 90.82 | 99.48 | **99.85** | 75.67 | 72.28 | 66.24 | 61.20 | 61.56 | 68.83 | 57.34 | 53.51 | 72.62 | 79.95 |
| FreDect | 99.99 | 88.98 | 93.62 | 84.78 | 99.49 | 82.84 | 82.54 | 55.85 | 61.77 | 52.92 | 46.09 | 37.83 | 37.76 | 85.10 | 39.58 | 38.20 | 49.45 | 66.87 |
| Fusing | **100.00** | 99.50 | 90.70 | 95.50 | 99.80 | 88.30 | 99.60 | 93.30 | 94.10 | 77.50 | 70.00 | 65.40 | 65.70 | 75.60 | 64.60 | 68.12 | 79.41 | 83.95 |
| LNP | 99.89 | 98.60 | 84.32 | 92.83 | **100.00** | 78.85 | 99.59 | 91.45 | 94.20 | 88.86 | 76.86 | 94.31 | 93.92 | 87.35 | 92.38 | 96.14 | 87.75 | 91.29 |
| LGrad | **100.00** | 98.31 | 92.93 | 95.01 | **100.00** | 95.43 | 97.89 | 57.99 | 72.95 | 80.42 | 71.86 | 62.37 | 62.85 | 77.47 | 62.48 | 82.55 | 80.03 | 81.80 |
| UnivFD | 99.08 | 91.74 | 75.25 | 80.56 | 99.34 | 72.15 | 88.29 | 60.13 | 85.84 | 78.35 | 61.86 | 49.87 | 49.52 | 54.57 | 55.38 | 74.48 | 67.59 | 89.73 |
| DIRE-G | 58.79 | 56.68 | 46.91 | 50.03 | 40.64 | 47.34 | 58.03 | 59.02 | **99.79** | 99.54 | 97.32 | 98.61 | 98.83 | 98.98 | 98.37 | 99.71 | 53.97 | 72.38 |
| DIRE-D | **100.00** | 97.56 | 99.27 | 99.80 | 99.37 | 99.98 | 97.90 | 96.73 | 86.81 | 83.81 | 74.00 | 86.14 | 85.84 | 96.53 | 91.07 | 63.04 | 99.10 | 76.92 |
| PatchCraft | **100.00** | 98.96 | 99.42 | 85.26 | **100.00** | 81.33 | 97.74 | 95.26 | 93.40 | 94.04 | 96.48 | 99.06 | 99.06 | 96.26 | 97.54 | 99.56 | 99.89 | 96.07 |
| NPR | **100.00** | 99.81 | 87.87 | 98.55 | 99.90 | 85.57 | 99.90 | 65.38 | 74.61 | 85.73 | 85.41 | 84.02 | 84.67 | 81.20 | 80.51 | 76.72 | **100.00** | 87.64 |
| AIDE | **100.00** | **99.99** | 94.44 | 99.89 | 99.99 | 97.69 | **99.96** | 99.27 | 98.77 | 98.94 | 88.13 | 98.26 | 98.20 | **99.27** | 98.62 | 99.41 | 99.31 | 98.25 |
| Effort | **100.00** | 99.40 | **99.97** | **100.00** | **100.00** | **100.00** | 98.85 | 99.59 | 92.64 | 87.80 | 53.19 | 67.20 | 66.98 | 92.84 | 75.15 | 51.65 | 60.45 | 85.04 |
| OmniAID-Mirage | 99.97 | 99.71 | 95.19 | 99.48 | **100.00** | 99.35 | 99.55 | 96.71 | 89.21 | **99.83** | **99.81** | **99.99** | **99.99** | 97.55 | **99.98** | **99.94** | 99.97 | **98.60** |

*Table 9.* Performance comparison on the DRCT-2M. We report detection Accuracy (ACC %). All baselines are trained on SD v1.4 , whereas our OmniAID-Mirage is trained on the Mirage-Train. The **best** and second-best results are marked in bold and underline, respectively.

| Method | SD Variants | | | | | Turbo Variants | | | LCM Variants | | ControlNet Variants | | | DR Variants | | | Mean |
|---|---|---|---|---|---|---|---|---|---|---|---|---|---|---|---|---|---|
| | LDM | SDv1.4 | SDv1.5 | SDv2 | SDXL | SDXL-Refiner | SD-Turbo | SDXL-Turbo | LCM-SDv1.5 | LCM-SDXL | SDv1-Ctrl | SDv2-Ctrl | SDXL-Ctrl | SDv1-DR | SDv2-DR | SDXL-DR | |
| CNNSpot | 99.87 | 99.91 | 99.90 | **97.55** | 66.25 | 86.55 | 86.15 | 72.42 | 98.26 | 61.72 | 97.96 | 85.89 | 82.84 | 60.93 | 51.41 | 50.28 | 81.12 |
| F3Net | 99.85 | 99.78 | 99.79 | 88.66 | 55.85 | 87.37 | 68.29 | 63.66 | 97.39 | 54.98 | 97.98 | 72.39 | 81.99 | 65.42 | 50.39 | 50.27 | 77.13 |
| CLIP/RN50 | 99.00 | 99.99 | 99.96 | 94.61 | 62.08 | 91.43 | 83.57 | 64.40 | 98.97 | 57.43 | 99.74 | 80.69 | 82.03 | 65.83 | 50.67 | 50.47 | 80.05 |
| GramNet | 99.40 | 99.01 | 98.84 | 95.30 | 62.63 | 80.68 | 71.19 | 69.32 | 93.05 | 57.02 | 89.97 | 75.55 | 82.68 | 51.23 | 50.01 | 50.08 | 76.62 |
| De-fake | 92.10 | 99.53 | 99.51 | 89.65 | 64.02 | 69.24 | **92.00** | 93.93 | 99.13 | 70.89 | 58.98 | 62.34 | 66.66 | 50.12 | 50.16 | 50.00 | 75.52 |
| Conv-B | **99.97** | 100.0 | 99.97 | 95.84 | 64.44 | 82.00 | 80.82 | 60.75 | 99.27 | 62.33 | 99.80 | 83.40 | 73.28 | 61.65 | 51.79 | 50.43 | 79.11 |
| UnivFD | 98.30 | 96.22 | 96.33 | 93.83 | 91.01 | 93.91 | 86.38 | 85.92 | 90.44 | 88.99 | 90.41 | 81.06 | 89.06 | 51.96 | 51.03 | 50.46 | 83.46 |
| DIRE | 98.19 | 99.94 | 99.96 | 68.16 | 53.84 | 71.93 | 58.87 | 54.35 | **99.78** | 59.73 | 99.65 | 64.20 | 59.13 | 51.99 | 50.04 | 49.97 | 71.23 |
| DRCT | 99.91 | 99.90 | 99.90 | 96.32 | 83.87 | 85.63 | 91.88 | 70.04 | 99.66 | 78.76 | **99.90** | **95.01** | 81.21 | **99.90** | 95.40 | 75.39 | 90.79 |
| OmniAID-Mirage | 90.62 | 98.45 | 98.43 | 96.60 | 92.02 | **97.33** | 82.34 | 71.60 | 97.86 | **98.61** | 94.19 | 72.51 | 84.20 | 98.88 | **98.82** | **98.15** | **91.91** |

*Table 10.* Performance comparison on the DRCT-2M. We report detection F1 (%). All baselines are trained on SD v1.4 , whereas our OmniAID-Mirage is trained on the Mirage-Train. The **best** and second-best results are marked in bold and underline, respectively.

| Method | SD Variants | | | | | Turbo Variants | | | LCM Variants | | ControlNet Variants | | | DR Variants | | | Mean |
|---|---|---|---|---|---|---|---|---|---|---|---|---|---|---|---|---|---|
| | LDM | SDv1.4 | SDv1.5 | SDv2 | SDXL | SDXL-Refiner | SD-Turbo | SDXL-Turbo | LCM-SDv1.5 | LCM-SDXL | SDv1-Ctrl | SDv2-Ctrl | SDXL-Ctrl | SDv1-DR | SDv2-DR | SDXL-DR | |
| CNNSpot | 99.87 | 99.91 | 99.90 | **97.49** | 49.13 | 84.48 | 83.94 | 61.97 | 98.23 | 38.08 | 97.92 | 83.59 | 79.31 | 35.98 | 05.67 | 01.31 | 69.80 |
| F3Net | 99.85 | 99.78 | 99.79 | 84.24 | 21.20 | 85.57 | 53.69 | 43.08 | 97.32 | 18.38 | 97.94 | 61.95 | 78.08 | 47.29 | 01.90 | 01.43 | 61.97 |
| CLIP/RN50 | 99.99 | 99.99 | 99.96 | 94.30 | 38.94 | 90.63 | 80.34 | 47.74 | 98.96 | 25.90 | 99.97 | 76.07 | 78.10 | 48.11 | 02.68 | 01.90 | 67.72 |
| GramNet | 99.40 | 99.01 | 98.83 | 95.10 | 40.99 | 76.26 | 59.92 | 56.18 | 92.59 | 25.54 | 88.94 | 67.93 | 79.23 | 06.09 | 01.42 | 01.69 | 61.82 |
| De-fake | 91.45 | 99.53 | 99.51 | 88.50 | 44.10 | 55.79 | **91.34** | 93.56 | 99.13 | 59.13 | 30.85 | 39.92 | 50.24 | 01.15 | 01.31 | 00.68 | 59.14 |
| Conv-B | 99.97 | 100.0 | 99.97 | 95.66 | 44.82 | 78.05 | 76.27 | 35.39 | 99.26 | 39.56 | 99.80 | 80.10 | 63.54 | 37.79 | 06.91 | 01.63 | 66.17 |
| UnivFD | 98.29 | 96.11 | 96.22 | 93.48 | 90.21 | 93.57 | 84.39 | 83.78 | 89.53 | 87.75 | 89.49 | 76.88 | 87.83 | 09.01 | 05.63 | 03.47 | 74.10 |
| DIRE | 98.16 | 99.94 | 99.96 | 53.33 | 14.36 | 61.01 | 30.21 | 16.10 | **99.78** | 32.65 | 99.65 | 44.29 | 30.95 | 07.76 | 00.28 | 00.00 | 49.28 |
| DRCT | 99.91 | 99.90 | 99.90 | 96.19 | 80.81 | 83.25 | 91.18 | 57.33 | 99.66 | 73.09 | 99.90 | 94.76 | 76.91 | **99.90** | 95.19 | 67.43 | 88.46 |
| OmniAID-Mirage | 89.90 | 98.46 | 98.44 | 96.56 | 91.53 | **97.32** | 79.12 | 61.53 | 97.86 | **98.62** | 93.97 | 63.21 | 81.72 | 98.89 | **98.83** | **98.16** | **90.26** |

## E.2. Comparison on DRCT-2M

The DRCT-2M benchmark serves as a rigorous testbed for generalization, incorporating diverse diffusion architectures (e.g., SDXL, Turbo, LCM) and challenging reconstruction-based attacks (DR). As detailed in Tables 9 to 11, OmniAID-Mirage demonstrates superior robustness compared to the baselines. While traditional methods struggle with modern generators and fail significantly on DR variants (where most methods exhibit an FNR > 99%), OmniAID maintains robust performance, achieving 92.02% accuracy on SDXL and 98.15% accuracy on SDXL-DR. Overall, our method establishes a new state-of-the-art, securing the highest mean accuracy and F1-score alongside the lowest FNR. This validates that our Universal Artifact Expert effectively captures intrinsic, content-agnostic inconsistencies that persist across varying architectures and obfuscation techniques.

*Table 11.* Performance comparison on the DRCT-2M. We report detection False Negative Rate (FNR, %). All baselines are trained on SD v1.4 , whereas our OmniAID-Mirage is trained on the Mirage-Train. The **best** and second-best results are marked in bold and underline, respectively.

| Method | SD Variants | | | | | Turbo Variants | | | LCM Variants | | ControlNet Variants | | | DR Variants | | | *Mean* |
|---|---|---|---|---|---|---|---|---|---|---|---|---|---|---|---|---|---|
| | LDM | SDv1.4 | SDv1.5 | SDv2 | SDXL | SDXL-Refiner | SD-Turbo | SDXL-Turbo | LCM-SDv1.5 | LCM-SDXL | SDv1-Ctrl | SDv2-Ctrl | SDXL-Ctrl | SDv1-DR | SDv2-DR | SDXL-DR | |
| CNNSpot | 00.16 | 00.08 | 00.10 | 04.80 | 67.40 | 26.80 | 27.60 | 55.06 | 03.38 | 76.46 | 03.98 | 28.12 | 34.22 | 78.04 | 97.08 | 99.34 | 37.66 |
| F3Net | 00.12 | 00.26 | 00.24 | 22.50 | 88.12 | 25.08 | 63.24 | 72.50 | 05.04 | 89.86 | 03.86 | 55.04 | 35.84 | 68.98 | 99.04 | 99.28 | 45.56 |
| CLIP/RN50 | **00.00** | **00.00** | 00.06 | 10.76 | 75.82 | 17.12 | 32.84 | 71.18 | 02.04 | 85.12 | 00.50 | 38.60 | 35.92 | 68.32 | 98.64 | 99.04 | 39.75 |
| GramNet | 00.50 | 01.28 | 01.62 | 08.70 | 74.04 | 37.94 | 56.92 | 60.66 | 13.20 | 85.26 | 19.36 | 48.20 | 33.94 | 96.84 | 99.28 | 99.14 | 46.06 |
| De-fake | 15.46 | 00.60 | 00.64 | 20.36 | 71.62 | 61.18 | **15.66** | **11.80** | 01.40 | 57.88 | 81.70 | 74.98 | 66.34 | 99.42 | 99.34 | 99.66 | 48.63 |
| Conv-B | 00.06 | **00.00** | 00.06 | 08.32 | 71.12 | 36.00 | 38.36 | 78.50 | 01.46 | 75.34 | 00.40 | 33.20 | 53.44 | 76.70 | 96.42 | 99.18 | 41.79 |
| UnivFD | 02.54 | 06.70 | 06.48 | 11.48 | 17.12 | 11.32 | 26.38 | 27.30 | 18.26 | 21.16 | 18.32 | 37.02 | 21.02 | 95.24 | 97.08 | 98.22 | 32.23 |
| DIRE | 03.56 | 00.06 | **00.02** | 63.62 | 92.26 | 56.08 | 82.20 | 91.24 | **00.38** | 80.48 | 00.64 | 71.54 | 81.68 | 95.96 | 99.86 | 100.0 | 57.47 |
| DRCT | **00.00** | 00.02 | **00.02** | 07.18 | 32.08 | 28.56 | 16.06 | 59.74 | 00.50 | 42.30 | **00.02** | **00.98** | 37.40 | **00.02** | 09.02 | 49.04 | 17.68 |
| OmniAID-Mirage | 16.54 | 00.88 | 00.92 | **04.58** | **13.74** | **03.12** | 33.10 | 54.58 | 02.06 | **00.56** | 09.40 | 52.76 | 29.38 | 00.02 | **00.14** | **01.48** | **13.95** |

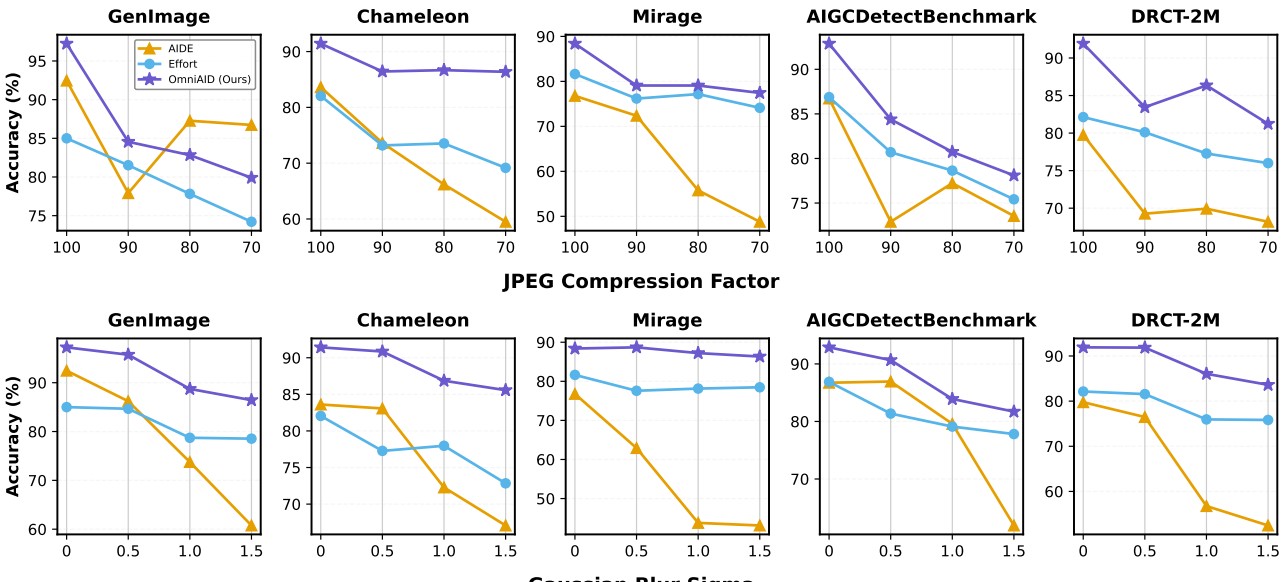

*Figure 8.* Robustness evaluation against post-processing perturbations. All models are trained on the Mirage-Train. The top row displays the performance degradation under varying JPEG compression factors (100, 90, 80, 70), while the bottom row shows the impact of Gaussian blur with increasing sigma values (0, 0.5, 1.0, 1.5). OmniAID (purple star) demonstrates superior stability compared to AIDE (orange triangle) and Effort (blue circle) across all five datasets.

# F. Robustness Evaluation

Real-world images frequently undergo post-processing operations such as compression or blurring. To rigorously assess the intrinsic robustness of our method against such corruptions, we trained all competing models exclusively on the **Mirage-Train** dataset without applying any data augmentation (other than standard resizing and normalization). This experimental setup ensures that the observed stability stems from the method's inherent representational capability rather than invariance induced by augmentation.

The evaluation results across five benchmarks, subject to varying degrees of JPEG compression and Gaussian blur, are presented in Figure 8. As observed, OmniAID consistently demonstrates superior stability compared to AIDE and Effort. Under JPEG compression, while all methods experience performance degradation, OmniAID maintains the highest accuracy; notably, on GenImage, it significantly outperforms Effort even at a quality factor of 70. This advantage becomes even more pronounced under Gaussian blur. Crucially, AIDE suffers a catastrophic performance collapse on datasets such as Mirage and DRCT-2M (dropping to ∼43% and ∼52% at $\sigma = 1.5$, respectively), whereas OmniAID remains remarkably stable, maintaining over 83% accuracy. This indicates that OmniAID captures robust, generalized features that are resilient to low-level signal corruption, rather than overfitting to fragile high-frequency artifacts.

*Table 12.* Ablation study analyzing the impact of different training protocols and artifact expert configurations. We compare our proposed two-stage strategy with a standard end-to-end baseline and a variant with a routable artifact expert. All models are trained on the GenImage-SD v1.4.

| Training Strategy | GenImage | Chameleon | Mirage |
|---|---|---|---|
| Standard End-to-End | 86.57 | 71.52 | 42.29 |
| Unfixed Artifact Expert | 89.70 | 70.23 | 49.73 |
| Our Two Stage | **95.94** | **77.35** | **51.10** |

*Table 13.* Effect of expert initialization. Both variants use the same OmniAID architecture and are trained on GenImage-SD v1.4.

| Initialization | GenImage | Chameleon | Mirage |
|---|---|---|---|
| LoRA | **96.03** | 76.79 | 49.77 |
| Effort-style SVD | 95.94 | **77.35** | **51.10** |

*Table 14.* Sensitivity to semantic expert training order. All models are trained on GenImage-SD v1.4.

| Order | GenImage | Chameleon | Mirage |
|---|---|---|---|
| Current | **95.94** | 77.35 | **51.10** |
| Reverse order | 95.75 | **78.28** | 48.49 |

# G. Additional Ablation Studies

We conduct comprehensive ablation studies to validate the individual contributions of each component within the OmniAID framework. Unless otherwise specified, all ablation experiments are performed on the GenImage-SDv1.4 training set for efficiency.

## G.1. Effect of Training Strategy

We empirically investigate the impact of our training paradigm and expert configuration in Table 12.

- **Necessity of Two-Stage Training:** We compare our approach against a "Standard End-to-End" baseline, in which all experts and the router are optimized jointly in a single stage. The substantial performance decline (from 95.94% to 86.57% on GenImage) indicates that joint optimization induces optimization interference, thereby hindering experts from achieving distinct specialization. This confirms that our Stage 1 (Expert Specialization) is a prerequisite for the router to effectively learn semantic-aware dispatching in Stage 2.

- **Fixed vs. Routable Artifact Expert:** We further evaluate an "Unfixed Artifact Expert" variant, wherein the artifact expert participates in the routing process alongside the semantic experts (i.e., it is routable rather than globally active). Although this variant outperforms the end-to-end baseline, it still underperforms compared to our proposed fixed design (89.70% vs. 95.94%). This result corroborates our hypothesis that generator artifacts are content-agnostic and ubiquitous; consequently, the Artifact Expert should serve as a *fixed, universal anchor* active for all inputs, rather than competing with semantic experts during the routing decision.

## G.2. Initialization and Training Order

To further disentangle the source of gains, we analyze expert initialization and sequential training order. First, Table 13 compares our Effort-style SVD initialization with a standard LoRA initialization under the same OmniAID architecture. Both variants remain competitive, indicating that the main improvement stems from the proposed dual-decoupling expert architecture rather than from a specific low-rank initializer alone. SVD initialization is adopted because it provides slightly stronger OOD performance. Second, Table 14 evaluates whether the sequential expert specialization order dominates performance. Reversing the order causes only moderate changes, suggesting that the orthogonality term acts as a soft diversity regularizer rather than a hard exclusion constraint that prevents later experts from learning complementary information.

*Table 15.* Component-wise ablation study quantifying the contribution of each optimization objective. We report the performance impact of removing Orthogonality ($\mathcal{L}_{\text{orth}}$), Gating Supervision ($\mathcal{L}_{\text{gating}}$), and Load Balancing ($\mathcal{L}_{\text{balance}}$) terms. All models are trained on the GenImage-SD v1.4.

| | Loss | | GenImage | Chameleon | Mirage-Test |
|---|---|---|---|---|---|
| $\mathcal{L}_{\text{orth}}$ | $\mathcal{L}_{\text{gating}}$ | $\mathcal{L}_{\text{balance}}$ | | | |
| $\times$ | $\times$ | $\times$ | 91.72 | 75.86 | 45.15 |
| $\checkmark$ | $\times$ | $\times$ | 94.16 | 77.66 | 49.41 |
| $\times$ | $\checkmark$ | $\times$ | 92.97 | 79.60 | 47.25 |
| $\times$ | $\times$ | $\checkmark$ | 92.85 | **79.96** | 47.19 |
| $\checkmark$ | $\checkmark$ | $\times$ | 94.03 | 77.32 | 51.03 |
| $\checkmark$ | $\times$ | $\checkmark$ | 93.98 | 78.21 | 49.90 |
| $\times$ | $\checkmark$ | $\checkmark$ | 92.97 | 79.64 | 47.26 |
| $\checkmark$ | $\checkmark$ | $\checkmark$ | **95.94** | 77.35 | **51.10** |

## G.3. Effect of Training Objectives

We systematically evaluate the contribution of each loss term by measuring the performance degradation incurred upon its removal, as summarized in Table 15.

- **Impact of Orthogonality ($\mathcal{L}_{orth}$):** This component constitutes a fundamental pillar of our framework. Comparing the full model (Row 8) with the variant lacking orthogonality (Row 7), we observe a substantial performance decline on both GenImage ($95.94\% \rightarrow 92.97\%$) and Mirage-Test ($51.10\% \rightarrow 47.26\%$). This confirms that without explicit orthogonality constraints, the residual experts fail to learn distinct, decoupled representations, leading to feature redundancy and compromised generalization.

- **Impact of Gating Supervision ($\mathcal{L}_{gating}$):** The exclusion of gating supervision (Row 6) results in a marked reduction in accuracy. Given that our experts are pre-specialized in Stage 1, $\mathcal{L}_{gating}$ proves critical for aligning the router's dispatching logic with the experts' intrinsic semantics. **This is further corroborated by our qualitative analysis in Figure 9**, which reveals that the absence of $\mathcal{L}_{gating}$ results in erratic and semantically incoherent routing assignments (e.g., assigning "Human" images to the "Object" expert). This demonstrates that the router fails to spontaneously acquire accurate semantic mappings without explicit guidance.

- **Impact of Load Balancing ($\mathcal{L}_{balance}$):** Comparing Row 5 with the full model, the incorporation of the load balancing loss yields a performance gain of nearly 2% on GenImage ($94.03\% \rightarrow 95.94\%$). This validates its efficacy in preventing "expert collapse," ensuring that the model leverages the full capacity of the expert pool rather than overfitting to a single dominant expert.

## G.4. Influence of Loss Coefficients

We perform a detailed sensitivity analysis on the weighting coefficients $\lambda_1$, $\lambda_2$, and $\lambda_3$ to strictly determine the optimal configuration for our multi-objective optimization. The results are summarized in Table 16.

- **Orthogonality Coefficient ($\lambda_1$):** This coefficient governs the strength of the orthogonality constraint, which enforces separation not only between the principal and residual subspaces but also mutually among different expert subspaces. As observed, setting $\lambda_1 = 0.1$ imposes an overly aggressive constraint. Although this configuration significantly enhances OOD generalization on Mirage, it precipitates a severe degradation in in-domain accuracy. We hypothesize that enforcing such rigid orthogonality between semantic domains disrupts the intrinsic feature correlations required for effective classification, leading to a drastic performance decline on GenImage ($88.28\%$). Consequently, we reject this setting to preserve the discriminative integrity of the source domain, prioritizing a balanced configuration that secures robust generalization without compromising fundamental classification capability. Conversely, for the GenImage subset, a too lenient $\lambda_1 = 0.001$ fails to prevent expert redundancy, resulting in suboptimal performance on the challenging Mirage test set. We find that $\lambda_1 = 0.01$ offers the best trade-off in this experimental setting, effectively decoupling expert roles without compromising feature integrity. (Note: For the large-scale Mirage-Train training, we relax this constraint to 0.001 as the increased data diversity naturally mitigates redundancy).

*Table 16.* Sensitivity analysis of the loss coefficients. We investigate the impact of varying the coefficients for Orthogonality ($\lambda_1$), Gating Supervision ($\lambda_2$), and Load Balancing ($\lambda_3$) on detection performance across different domains. All models are trained on the GenImage-SD v1.4.

| $\lambda_1$ | GenImage | Chameleon | Mirage | $\lambda_2$ | GenImage | Chameleon | Mirage | $\lambda_3$ | GenImage | Chameleon | Mirage |
|---|---|---|---|---|---|---|---|---|---|---|---|
| 0.001 | 94.96 | **78.50** | 49.03 | 0.01 | 93.99 | **77.55** | **51.41** | 0.01 | 94.16 | 77.32 | 48.44 |
| 0.01 | **95.94** | 77.35 | 51.10 | 0.1 | **95.94** | 77.35 | 51.10 | 0.1 | **95.94** | 77.35 | **51.10** |
| 0.1 | 88.28 | 61.45 | **58.69** | 1.0 | 95.94 | 77.32 | 51.03 | 1.0 | 95.93 | **77.55** | 48.73 |

*Table 17.* Ablation study investigating the impact of the Top-$K$ parameter in the OmniAID router. All models are trained on the Mirage-Train. "FPS (100 BS)" denotes the inference throughput (frames per second) measured on the Chameleon dataset with a batch size of 100 per GPU.

| Top-$K$ | GenImage | Chameleon | Mirage | AIGCDetectBenchmark | DRCT-2M | FPS (100 BS) |
|---|---|---|---|---|---|---|
| 1 | 97.11 | 90.54 | 87.01 | 92.64 | 91.84 | 201.38 |
| 2 | 97.24 | 91.42 | 88.39 | 92.88 | 91.91 | 191.99 |
| 3 | 97.27 | 91.53 | 88.49 | 92.88 | 92.11 | 182.59 |
| 4 | **97.29** | **91.58** | **88.62** | **92.90** | 92.12 | 170.78 |
| 5 | 97.28 | 91.56 | 88.56 | 92.89 | 92.13 | 165.02 |

- **Gating Coefficient ($\lambda_2$):** The model exhibits robustness to variations in the gating supervision weight. While $\lambda_2 = 0.01$ achieves marginally higher OOD scores, it compromises in-domain accuracy (93.99%). We select $\lambda_2 = 0.1$ as the optimal point, maximizing GenImage performance (95.94%) with negligible trade-offs on OOD benchmarks, ensuring reliable semantic routing.

- **Balance Coefficient ($\lambda_3$):** Proper magnitude for the load balancing term is crucial. On the GenImage subset, a small $\lambda_3$ (0.01) is insufficient to counteract the "winner-takes-all" tendency, resulting in lower generalization performance on Mirage (48.44%). Increasing $\lambda_3$ to 0.1 significantly improves robustness (+2.66% on Mirage) by enforcing a more equitable expert utilization. Thus, we adopt $\lambda_3 = 0.1$ as the optimal setting for this scale. (Note: For the large-scale Mirage-Train training, the inherent data diversity naturally encourages expert utilization; therefore, we relax this constraint to $\lambda_3 = 0.001$ to avoid over-regularization during scaling).

## G.5. Influence of Top-$K$

We explore the optimal number of active experts $K$ during inference, with results presented in Table 17 (trained on Mirage-Train) and Table 18 (trained on GenImage-SD v1.4). We observe a distinct correlation between the training data complexity and the optimal $K$.

- **Single Expert for Homogeneous Data:** As shown in Table 18, when the model is trained on the relatively homogeneous GenImage dataset, setting $K = 1$ yields the best generalization performance (e.g., 51.10% on Mirage vs. 50.70% with $K = 2$). Activating more experts ($K = 2$) slightly improves source domain accuracy (95.97% vs. 95.94%) but degrades performance on unseen domains, indicating a tendency towards overfitting.

- **Expert Collaboration for Diverse Data:** Conversely, for the highly diverse Mirage-Train dataset (Table 17), relying on a single expert is insufficient. Increasing $K$ from 1 to 2 brings significant gains across all benchmarks (e.g., +1.38% on Mirage and +0.88% on Chameleon). This suggests that complex, real-world forgeries require the collaboration of multiple experts to capture complementary semantic artifacts.

- **Efficiency Trade-off:** In Table 17, while $K = 4$ achieves the highest accuracy, the marginal gain over $K = 2$ is minimal (e.g., +0.23% on Mirage) compared to the drop in inference throughput (∼21 FPS loss). Therefore, we adopt $K = 2$ as the default setting for our final Mirage-trained model to balance robustness and efficiency.

## G.6. Influence of Expert Rank ($r$)

The rank $r$ controls the capacity of our residual experts. We analyze its impact on the balance between fitting and generalization in Table 19.

*Table 18.* Ablation study investigating the impact of the Top-K parameter in the OmniAID router. Models are trained on the GenImage-SD v1.4. "FPS (100 BS)" denotes the inference throughput (frames per second) measured on the Chameleon dataset with a batch size of 100 per GPU.

| Top-$K$ | GenImage | Chameleon | Mirage | FPS (100 BS) |
|---|---|---|---|---|
| 1 | 95.94 | **77.35** | **51.10** | 207.68 |
| 2 | **95.97** | 77.24 | 50.70 | 198.22 |

*Table 19.* Sensitivity analysis of the expert adapter rank $r$. We evaluate the trade-off between model capacity (reflected by GenImage performance) and generalization robustness (reflected by Chameleon and Mirage). All models are trained on the GenImage-SD v1.4.

| r | GenImage | Chameleon | Mirage |
|---|---|---|---|
| 1 | 91.91 | 69.96 | 46.94 |
| 2 | 94.86 | 76.89 | 50.61 |
| 4 | 95.94 | **77.35** | **51.10** |
| 8 | **96.42** | 76.23 | 45.31 |
| 16 | 96.06 | 72.41 | 42.59 |

- **Capacity vs. Overfitting:** We observe a distinct bias-variance trade-off. While increasing the rank to $r = 8$ yields the highest accuracy on the source domain (GenImage: 96.42%), it leads to a performance decline on the unseen Chameleon and Mirage datasets. This indicates that excessive capacity encourages the model to overfit to source-specific artifacts rather than learning generalizable forgery traces.

- **Optimal Selection:** Conversely, lower ranks ($r \in \{1, 2\}$) suffer from underfitting due to insufficient representational capacity. The setting of $r = 4$ provides the optimal balance for the GenImage subset, achieving the best performance on both OOD benchmarks (Chameleon: 77.35%, Mirage: 51.10%) while maintaining competitive in-domain accuracy. Thus, we adopt $r = 4$ as the default for these ablation studies. However, for the model trained on the large-scale Mirage-Train, the demand for representational capacity is higher. Consequently, we scale the rank to $r = 8$ in our training for OmniAID-Mirage; this increased capacity allows for a more comprehensive capture of the artifact spectrum, while the larger data scale naturally mitigates the overfitting risks observed in the smaller dataset.

### G.7. Impact of Different ViT Backbones

We analyze the influence of the visual encoder's architecture and resolution in Table 20.

- **Model Scale:** Scaling up the model capacity from ViT-B to ViT-L yields a clear performance improvement (e.g., $80.44\% \rightarrow 89.07\%$ on GenImage). This indicates that the stronger semantic representation capabilities of larger foundational models are inherently beneficial for the detection task.

- **Impact of Resolution:** Comparing ViT-L/14 ($224 \times 224$ input) with ViT-L/14@336px, we observe a substantial gain across all metrics (e.g., $+6.87\%$ on GenImage). Since the model architecture remains identical, this performance gap strongly suggests that downsampling to lower resolutions discards critical discriminative information. The higher input resolution of 336px likely retains more fine-grained visual details, which enables the experts to capture subtler traces necessary for robust detection.

## H. Computational Cost

As presented in Table 21, Effort emerges as the most lightweight method, attaining the highest FPS (665.90) due to its minimal parameter updates. However, this efficiency significantly compromises generalization, particularly on challenging datasets such as Mirage (43.03%). Conversely, AIDE is computationally intensive, exhibiting the lowest inference throughput (55.36 FPS) and the highest training cost, yet it fails to deliver competitive performance on unseen domains. OmniAID achieves an optimal trade-off between efficiency and effectiveness. Relative to AIDE, it reduces training duration by $\sim25\%$ and accelerates inference by nearly $4\times$. Notably, although OmniAID implements an MoE architecture, it operates within the residual space of SVD decomposition; this design utilizes only 2.43M learnable parameters. We acknowledge, however,

*Table 20.* Ablation study investigating the impact of visual encoder architectures. All models are trained on the GenImage-SD v1.4.

| Backbone | GenImage | Chameleon | Mirage |
|---|---|---|---|
| ViT-B/32 | 80.44 | 72.00 | 48.87 |
| ViT-B/16 | 83.27 | 66.78 | 45.17 |
| ViT-L/14 | 89.07 | 75.51 | 47.25 |
| ViT-L/14@336px | **95.94** | **77.35** | **51.10** |

*Table 21.* Comparison of computational cost and detection performance. All models are trained on the GenImage-SD v1.4 dataset using 4 NVIDIA H200 GPUs. "Params (Learnable)" indicates the number of parameters updated during training. "FPS (100 BS)" denotes the inference throughput (frames per second) measured on the Chameleon dataset with a batch size of 100 per GPU.

| Method | Params | Params (Learnable) | GFLOPs | FPS (100 BS) | Train Time | GenImage | Chameleon | Mirage |
|---|---|---|---|---|---|---|---|---|
| AIDE | 897.83 M | 54.43 M | 225.69 G | 55.36 | 3.6 H | 86.88 | 62.60 | 31.25 |
| Effort | 303.38 M | 0.20 M | 51.95 G | 665.90 | 1.7 H | 91.10 | 62.06 | 43.03 |
| OmniAID | 508.78 M | 2.43 M | 291.34 G | 207.68 | 2.7 H | 95.94 | 77.35 | 51.10 |

that our method incurs higher total parameters and GFLOPs relative to the lightweight Effort. This is primarily attributed to the incorporation of an additional, frozen CLIP-ViT-L/14@336px encoder, which is employed to extract high-level semantic features for the router. While this visual encoding step introduces inherent computational overhead during inference, it constitutes a critical design choice that empowers our dynamic routing mechanism to effectively discriminate between diverse domains, yielding substantial performance gains (e.g., +15.29% on Chameleon over Effort).

## I. Additional Visualizations

To qualitatively validate the efficacy of our routing mechanism and underscore the necessity of gating supervision, we visualize the router's decision-making process in Figure 9.

**With $\mathcal{L}_{gating}$.** When trained with our full objective, the router exhibits distinct and semantically accurate activation patterns. As illustrated in the top row of Figure 9, input samples are correctly dispatched to their corresponding experts with high confidence (e.g., a "Human" image activates the Human Expert with a weight $> 0.9$). This confirms that the router successfully aligns visual features with the pre-defined expert specializations.

**Without $\mathcal{L}_{gating}$.** In the absence of explicit gating supervision, the router's behavior degrades significantly, as depicted in the bottom row of Figure 9.

- **Semantic Misalignment:** The router frequently assigns high weights to irrelevant domains. For instance, a distinct "Human" portrait is incorrectly routed to the "Object" expert with high confidence. This indicates that, without supervision, the router fails to establish a meaningful correspondence between input semantics and expert roles.

- **Unpredictability:** The weight distribution often becomes erratic or ambiguous, lacking the structured interpretability observed in the supervised model.

This visual evidence strongly corroborates our quantitative ablation studies, demonstrating that $\mathcal{L}_{gating}$ is indispensable. It ensures that the MoE architecture functions as a semantically organized system rather than an incoherent ensemble of random sub-networks.

## J. Samples in Mirage-Test

Figure 10 presents representative AI-generated samples from our Mirage-Test. This benchmark encompasses five distinct semantic categories: Human, Animal, Object, Scene, and Anime. Notably, the Anime category is broadly defined to include both Japanese anime and diverse cartoon styles. As illustrated, these samples exhibit exceptional visual fidelity, characterized by superior photorealism in natural domains and intricate detailing in stylized compositions.

## K. Limitation and Future Work

Despite setting a new state-of-the-art in robust AIGI detection, OmniAID exhibits certain limitations. First, our framework relies on a fixed taxonomy of semantic experts, which potentially constrains generalization to open-set domains that lie strictly outside these pre-defined categories. We plan to address this by leveraging our orthogonal subspace design for continual learning, thereby enabling the incremental integration of new semantic experts without catastrophic forgetting. Crucially, this extension would require optimizing only the new experts and updating the router. Second, the current semantic partition is coarse-grained and may be suboptimal; for instance, the "Anime" category intrinsically overlaps with "Human" and "Animal" semantics. Investigating more granular or data-driven subdivisions could further enhance generalization performance. Third, the explicit semantic router may introduce a potential attack surface if adversarial perturbations are designed to manipulate routing decisions. Our fixed artifact expert and Top-$K$ routing mitigate this risk by keeping category-agnostic evidence active and allowing multiple semantic experts to contribute, but robust routing under adaptive attacks remains an important direction for future work. Finally, while our Artifact Expert currently employs VAE-based reconstruction for computational efficiency, this proxy may not fully capture the entire spectrum of generative fingerprints. Future research could incorporate a broader array of heterogeneous VAEs or leverage direct generative model reconstruction to facilitate the learning of more comprehensive and robust artifact representations.

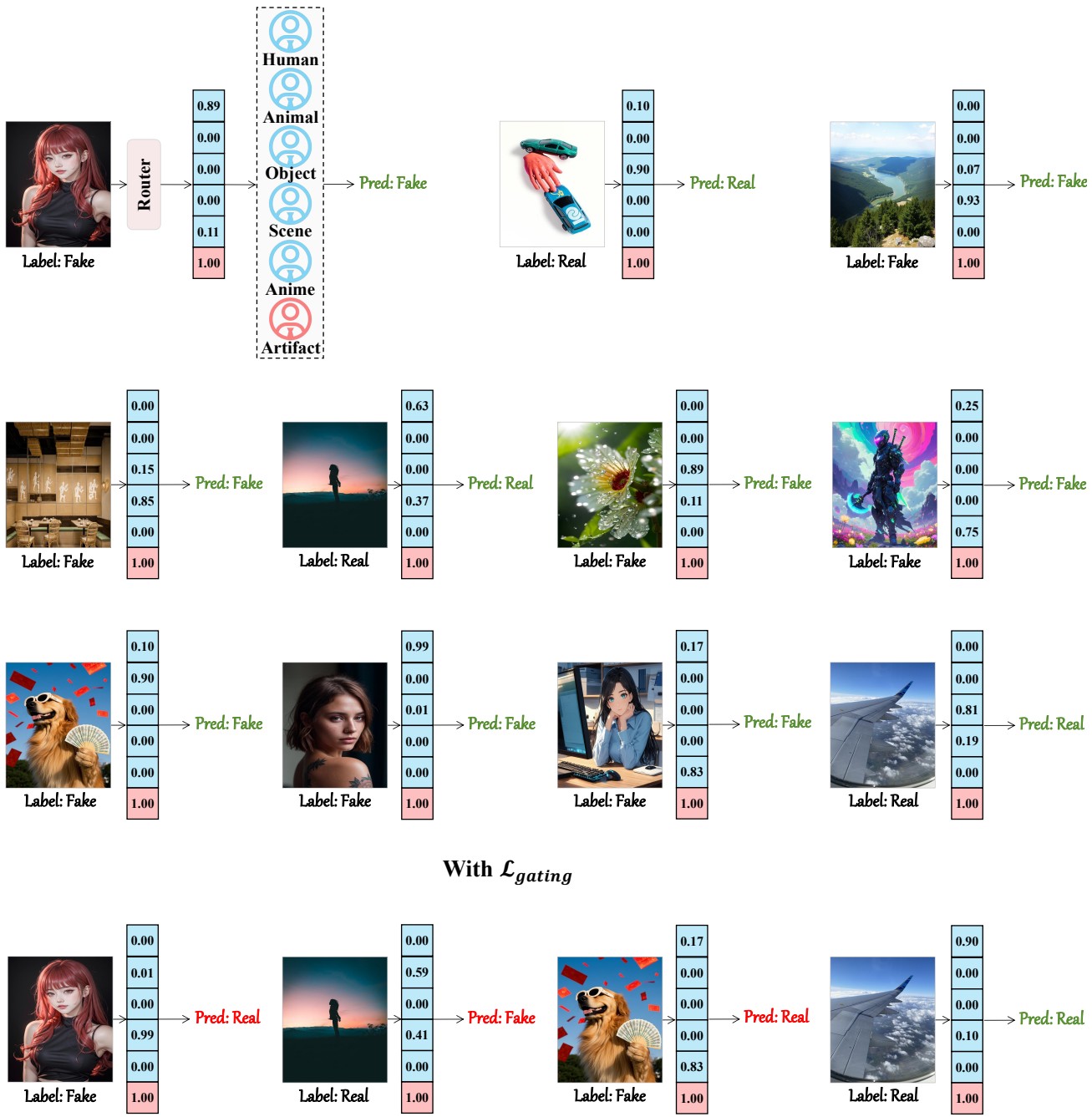

*Figure 9.* Visualization of the OmniAID routing mechanism. We compare the router's decision-making process with (top) and without (bottom) the proposed gating supervision loss $\mathcal{L}_{gating}$. As observed, $\mathcal{L}_{gating}$ ensures precise, semantically aligned expert selection, whereas removing it leads to chaotic, uninterpretable, and semantically mismatched routing behavior.

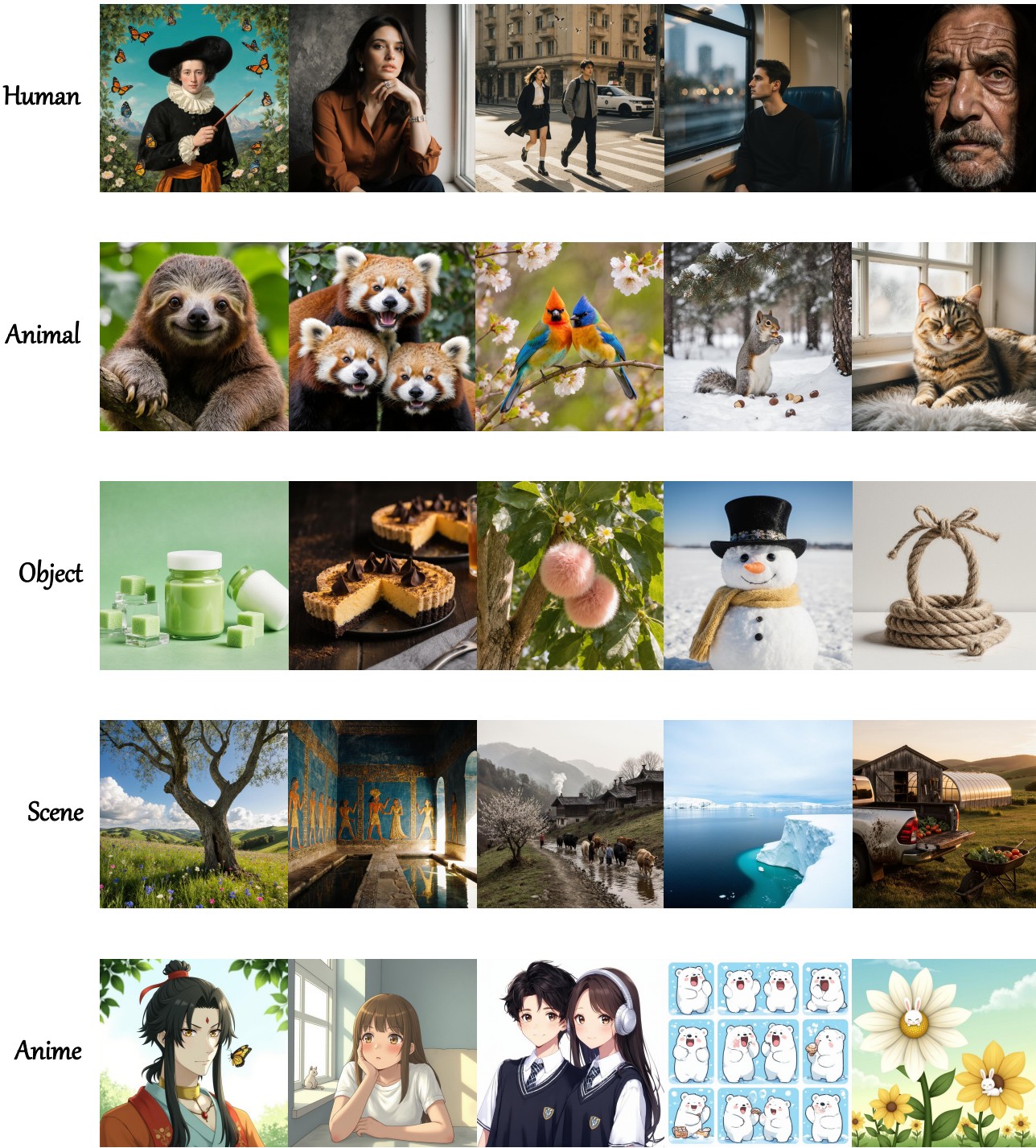

*Figure 10.* Random AI-generated samples from our Mirage-Test.

