# OpenReview forum: "OmniAID: Decoupling Semantic and Artifacts for Universal AI-Generated Image Detection in the Wild"
_ICML.cc/2026/Conference — ICML 2026 regular_

### Official Review · Reviewer_CmkP · 2026-03-04

**Soundness:** 3
**Presentation:** 3
**Significance:** 4
**Originality:** 3
**Overall Recommendation:** 4
**Confidence:** 4

**Summary:**

This paper introduces a method (OmniAID) and a dataset (Mirage) for AIGC detection. Specifically, OmniAID decouples semantics and artifacts, constructing a MoE framework to learn different semantics and low-level artifacts, and the proposed Mirage seems to be more challenging than existing datasets. The authors conduct thorough experiments on both existing benchmarks and Mirage, showing the strong performance of OmniAID.

**Compliance With Llm Reviewing Policy:**

Affirmed.

**Final Justification:**

My concerns are mostly addressed and I will keep my rating as weak accept. The core design (expert decoupling) does make sense and aligns with my own observations. Therefore, I maintain my score as weak accept.

**Key Questions For Authors:**

All my concerns and questions are listed in Weakness.

**Limitations:**

yes

**Strengths And Weaknesses:**

**Strength**
1. The paper introduces a thorough method and a dataset (Mirage) for AIGC detection. The performance in Table 5 indicates that Mirage may be a challenging benchmark for this field.
2. The authors conduct thorough experiments (on both traditional and the proposed datasets), which clearly demonstrate the efficacy of the proposed method.
3. The performance of the proposed method is impressive, e.g., OminiAID-Mirage achieves 91.4% Acc on Chameleon.
4. The paper is well written and easy to follow.


**Weakness**
1. My main concern is the novelty of the method: (1) The idea of decoupling semantics and artifacts is not new, which has already been explored in [1][2], (2) the artifact experts directly taking VAE reconstruction, which has been broadly adopted in recent works such as [3][4], and (3) the model training also appears to follow Effort[5]. As a result, the proposed method seems more like a method ensemble and engineering optimization rather than a principled finding or exploration.

2. What generative models does Mirage-Test contain?

3. Why does the Artifact Expert use VAE reconstructed data? In this regard, I have a few questions:

    3.1 How to define "Artifact" here? Or in other words, what does this branch actually learn?

    3.2 Can this generalize to GAN-based models?

    3.3 How does the choice of VAE affect the results?

    3.4 Why is this branch treated as a universal expert (i.e., assigned a weight of 1)?

4. How to determine N_s? How would a more fine-grained semantic categorization here affect the training efficiency and performance? It's better to have a further discussion.



[1] CO-SPY: Combining Semantic and Pixel Features to Detect Synthetic Images by AI. CVPR 2025

[2] AlignGemini: Generalizable AI-Generated Image Detection Through Task-Model Alignment

[3] Dual Data Alignment Makes AI-Generated Image Detector Easier Generalizable. NeurIPS 2025

[4] Aligned Datasets Improve Detection of Latent Diffusion-Generated Images. ICLR 2025

[5] Orthogonal Subspace Decomposition for Generalizable AI-Generated Image Detection. ICML 2025

---

> ### Author Rebuttal · Authors · 2026-03-30
>
> Thank you for the valuable feedback on our paper.
>
> ### Q1. On the novelty concern
> We do not claim novelty for any single component. The core contribution lies in a **unified dual-decoupling and routing architecture**. This is the first framework to explicitly perform **dual decoupling** (across semantic domains and between semantics and artifacts) via **two-stage optimization within a single foundation model**.
>
> **(1) vs. multi-branch/multi-model fusion (AlignGemini, CO-SPY).** These methods combine independent networks (e.g., VLM and CNN) at the feature level. OmniAID performs adaptation at the **parameter level**, using MoE within a single model's low-rank residual space. This avoids representation gaps and improves efficiency. Beyond this, the modeling perspective also differs. On the semantic side, these works rely on shared representations, while OmniAID introduces specialized semantic experts with dynamic routing, improving robustness under semantic shift. On the artifact side, these methods focus on narrow signals like pixel-level texture inconsistencies, whereas OmniAID learns content-agnostic representations via a universal expert trained with semantically aligned hard pairs, enabling broader generalization.
>
> **(2) vs. VAE reconstruction (DDA, AlignedForensics).**  In OmniAID, VAE reconstruction serves as a pluggable strategy for artifact data construction rather than the core contribution, and our framework naturally benefits from community advances in hard-sample construction (e.g., we encourage adopting stronger strategies such as DDA's multi-alignment to further boost performance). More importantly, prior methods enforce artifact supervision within a single detector yet overlook the fact that VFM-based representations remain semantically dominated, leaving the model's semantic capacity underutilized. OmniAID addresses this by explicitly decoupling artifact and semantic modeling into separate expert branches.
>
> **(3) vs. Effort.** Effort is a monolithic detector rather than an expert-based system. Our training appears similar because we adopt Effort's SVD decomposition as **one possible initialization strategy**. Importantly, this choice is replaceable. As shown below, Effort-style initialization yields slightly better performance, especially on OOD benchmarks, but standard LoRA still achieves competitive results:
>
> | Initialization | GenImage | Chameleon | Mirage-Test |
> |:---:|:---:|:---:|:---:|
> | LoRA | **96.03** | 76.79 | 49.77 |
> | Effort-style SVD | 95.94 | **77.35** | **51.10** |
>
> This suggests the primary gains stem from the **unified dual-decoupling and routing architecture** rather than any specific initialization or low-rank operator. Further ablations show this is not a simple component combination. Removing semantic experts (Table 5), making the artifact expert routable (Table 12), or removing two-stage training (Table 12) all cause clear drops.
>
> ### Q2. What generative models Mirage-Test contains
> Mirage-Test is composed of **SOTA generators specifically tuned for high fidelity and minimal artifacts**. Specifically, we sourced customized LoRA modules from Civitai and Liblib (communities where users train their own generators with custom data) to enhance open-source generators such as SD or Flux. These include DigiCam, Flux_xhs_v2, amateurphoto-v6-forcu, realistic_photography_v1, ultrarealisticFinetune_v4, among others.
>
> ### Q3. Why the Artifact Expert uses VAE-reconstructed data
> **(3.1) What is "artifact"?** Content-agnostic, low-level forensic traces from generation/reconstruction (e.g., resampling, quantization, upsampling inconsistencies). The branch learns these universal signals for real/fake discrimination.
>
> **(3.2) Can this generalize to GANs?** Yes. The artifact expert aims to learn a generalizable real/fake boundary via hard-sample training, not all generator-specific artifacts. OmniAID-Mirage achieves **98.60** mean AP on AIGCDetectBenchmark (largely GAN-based), validating this.
>
> **(3.3) How does VAE choice affect results?** Using a single VAE risks overfitting to one reconstruction pattern (e.g., training with only the SD v1.4 VAE drops Chameleon accuracy by 5.62%). We therefore employ multiple VAEs (TAESD, TAESDXL, SD v1.x to SD 3.5) to encourage learning shared artifact patterns. Regarding why we choose VAE reconstruction, a key rationale is that it offers significantly greater efficiency than full diffusion-based pipelines while maintaining comparable performance, as detailed in AlignedForensics.
>
> **(3.4) Why fixed contribution?** The artifact branch is a **global forensic anchor** providing always-available, content-agnostic evidence. Routing it would force competition with semantic experts. Appendix Table 12 confirms the unfixed/routable variant is significantly weaker.
>
> ### Q4. Expert number and finer-grained taxonomy
> Please refer to our response to **Reviewer NFFT, Q2**, where we provide a detailed discussion on this question.

---

> > ### Author Rebuttal · Reviewer_CmkP · 2026-04-02
> >
> > My concerns are mostly addressed and I will keep my rating as weak accept. The core design (expert decoupling) does make sense and aligns with my own observations. The discussions in the rebuttal are suggested to add to the later version, especially questions in W3.

---

> > > ### Author Response · Authors · 2026-04-04
> > >
> > > We sincerely thank the reviewer for the positive feedback and for recognizing the effectiveness of our core design. Following your suggestion, we will incorporate the detailed discussions from our rebuttal (specifically regarding the VAE-reconstruction strategy) into the final version of the manuscript to enhance clarity. We appreciate your thoughtful comments, which help improve the quality of our work.

---

### Official Review · Reviewer_QkGU · 2026-03-10

**Soundness:** 4
**Presentation:** 4
**Significance:** 4
**Originality:** 3
**Overall Recommendation:** 5
**Confidence:** 5

**Summary:**

This paper proposes OmniAID, a novel Mixture-of-Experts (MoE) framework designed for universal AI-generated image detection. The core methodology addresses the entanglement of features by explicitly decoupling domain-specific semantic flaws from content-agnostic universal artifacts. Furthermore, the authors introduce Mirage, a large-scale modern dataset, to provide a rigorous and realistic evaluation of detector robustness against contemporary, high-fidelity generative models.

**Compliance With Llm Reviewing Policy:**

Affirmed.

**Final Justification:**

The rebuttal solves my concerns, and I choose to maintain my rate

**Key Questions For Authors:**

Please address the concerns raised in the weaknesses. And given that this architecture exposes an explicit "semantic router", is it possible for a future attacker to trick the router into sending images to the wrong expert, thus bypassing detection? I know this may be beyond the scope of this paper, but adding a brief discussion on this adversarial threat model would greatly enhance the paper's completeness regarding real-world security.

**Limitations:**

yes

**Strengths And Weaknesses:**

Strengths
1. The motivation to decouple high-level semantic flaws from low-level generator artifacts using an orthogonal Mixture-of-Experts (MoE) architecture is intuitive and well-justified. The introduction of a universal artifact expert effectively addresses a critical blind spot in current detectors based on vision foundation models.
2. The two-stage training strategy (expert specialization via hard-sampling, followed by router training) provides an stable and practical method for optimizing the decoupled experts. Meanwhile, the theoretical analysis regarding gradient orthogonality further solidifies the mathematical foundation of this approach.
3. The paper is clearly structured and strictly evaluated. The introduction of the Mirage dataset provides a much-needed benchmark for modern "in-the-wild" threats. Combined with comprehensive experiments and compelling visualizations (e.g., t-SNE feature space and router gating weights), the empirical results strongly demonstrate that the proposed decoupling mechanism achieves its intended effect.


Weaknesses
1. The Mirage dataset is a significant contribution, but the paper lacks transparency regarding the construction of the synthetic images. Specifically, it does not detail the source strategy for the text prompts used to guide the T2I models (e.g., whether they are randomly sampled, manually curated, or based on existing datasets like MS-COCO). Clarifying this point is essential for the completeness of the paper.
2. The methodology relies on a specific two-stage training pipeline (expert hard-sampling followed by router tuning) to achieve the desired feature decoupling. Conducting ablation studies to demonstrate why a more straightforward approach (e.g., using a vanilla MoE, jointly training the router and experts from scratch) fails or leads to suboptimal decoupling is necessary to fully justify this design choice.
3. OmniAID introduces a Mixture-of-Experts architecture requiring multiple domain-specific experts and a dedicated semantic router. While this decoupling significantly improves detection accuracy, it inherently raises concerns regarding deployment costs. To comprehensively evaluate its real-world applicability, the authors should include a quantitative comparison of the inference overhead (e.g., FLOPs, total parameter count, and latency) against standard monolithic foundation models.

---

> ### Author Rebuttal · Authors · 2026-03-30
>
> Thank you for the valuable feedback on our paper.
>
> ### Q1. The transparency of prompt sourcing for Mirage
> We agree that the current manuscript does not provide sufficiently detailed descriptions of the dataset construction process. We now clarify the full pipeline.
>
> For Mirage, our prompt construction follows a principled, real-image-anchored approach. First, to ensure high-quality real images as the foundation, we source a large collection of high-resolution photographs from public photography platforms such as Pexels, supplemented by human-created digital and anime art from online communities. Second, we employ LLMs (e.g., Gemini-3) to annotate each real image with both a **content description** (caption) and a **semantic category label** (Human/Animal/Object/Scene/Anime). Third, we use these content descriptions directly as prompts to generate corresponding synthetic images via diverse T2I generators. This real-image-anchored prompting strategy offers two key advantages: (1) it promotes coarse-level **semantic alignment** between real and fake images within each category, mitigating distribution mismatch between the two classes that could introduce spurious shortcuts; and (2) it guarantees that the prompts reflect **naturally occurring visual content** rather than artificially constructed or random text, thereby enhancing the real-world validity of the training data. We will supplement this pipeline in the revised manuscript.
>
> ### Q2. Whether the two-stage training is really necessary
> This can be directly supported by our ablation results. The following table (from Appendix Table 12) compares the two training strategies:
> | Training Strategy | GenImage | Chameleon | Mirage-Test |
> |:---|:---:|:---:|:---:|
> | End-to-end (joint training) | 86.57 | 71.52 | 42.29 |
> | **Two-stage (Ours)** | **95.94** | **77.35** | **51.10** |
> | Δ | **+9.37** | **+5.83** | **+8.81** |
>
> The substantial performance gap shows that joint optimization is far less effective than our two-stage pipeline. We attribute this to a chicken-and-egg dependency: without pre-established experts, the router lacks meaningful signals for semantic dispatching; without proper routing, experts receive mixed-domain data that hinders specialization. Our two-stage design breaks this cycle by first training specialized experts, then learning the router on top of frozen, already-competent experts.
>
> Additionally, we compared an **Unfixed Artifact Expert** variant (Appendix Table 12), where the artifact branch participates in routing like the semantic experts. This variant is also clearly weaker than our proposed fixed design. These ablations directly justify our current training pipeline, and we will present these results more prominently in the revised manuscript.
>
> ### Q3. Inference overhead, parameters, FLOPs, and latency
> This is a reasonable concern. OmniAID is not a full ensemble of multiple independent large detectors. It reuses a single frozen backbone and only introduces low-rank residual experts; at inference time, only a small number of semantic experts are activated. And we actually have provided a comprehensive efficiency-accuracy comparison with other methods in Appendix Table 19 as shown below:
>
> | Method | Params | Params (Learnable) | GFLOPs | FPS (100 BS) | Train Time | GenImage | Chameleon | Mirage-Test |
> |:---|:---:|:---:|:---:|:---:|:---:|:---:|:---:|:---:|
> | AIDE | 897.83 M | 54.43 M | 225.69 G | 55.36 | 3.6 H | 86.88 | 62.60 | 31.25 |
> | Effort | 303.38 M | 0.20 M | 51.95 G | 665.90 | 1.7 H | 91.10 | 62.06 | 43.03 |
> | **OmniAID** | 508.78 M | 2.43 M | 291.34 G | 207.68 | 2.7 H | **95.94** | **77.35** | **51.10** |
>
> OmniAID achieves the best detection performance across all benchmarks while maintaining a favorable efficiency trade-off. Compared to AIDE, OmniAID reduces training time by 25% and accelerates inference by nearly 4×. Although OmniAID has higher total parameters and GFLOPs than the lightweight Effort, this overhead is primarily attributed to the additional frozen CLIP-ViT-L/14@336px encoder used for router feature extraction. Despite this moderate cost, the overall framework delivers substantial performance gains (+15.29% on Chameleon, +8.07% on Mirage-Test over Effort), demonstrating a favorable accuracy-efficiency trade-off.
>
> ### Q4. Whether the semantic router introduces a new attack surface
> We agree this is an important security consideration. Our design incorporates two mitigation mechanisms. First, the universal artifact expert remains **always active**, so the detector does not rely entirely on routing to any single semantic expert. Second, we employ **Top-K** routing, enabling multiple experts to jointly contribute on ambiguous inputs, thereby reducing system fragility. Adversarial attacks targeting the router represent a meaningful future threat model and we will discuss this in the revision.

---

> > ### Author Rebuttal · Reviewer_QkGU · 2026-04-02
> >
> > Thanks to the authors for their meticulous response, which has addressed all my concerns. I maintain my score and recommend acceptance of this paper. I also encourage other reviewers to consider raising their scores and recommending acceptance.

---

> > > ### Author Response · Authors · 2026-04-04
> > >
> > > We sincerely thank you for the positive feedback and for acknowledging our efforts. We appreciate your support for our work and your recommendation to other reviewers.

---

### Official Review · Reviewer_wGen · 2026-03-12

**Soundness:** 2
**Presentation:** 3
**Significance:** 2
**Originality:** 3
**Overall Recommendation:** 3
**Confidence:** 4

**Summary:**

This paper addresses the insufficient cross-domain and cross-generalization capabilities of current AIGI models and proposes a detection framework, OmmniAID. Considering that existing VFM-based detectors conflate high-level, content-dependent semantic flaws with low-level, content-agnostic generator artifacts, OmmniAID employs a hybrid expert architecture. It utilizes a set of Routable Specialized Semantic Experts to target domain-specific semantic flaws and a Fixed Universal Artifact Expert to capture content-agnostic fingerprints. Furthermore, to address the issue of outdated benchmark datasets, the authors constructed the Mirage dataset.

**Compliance With Llm Reviewing Policy:**

Affirmed.

**Final Justification:**

The authors' response discusses this practical aspect solely from an algorithmic perspective and still fails to directly address how to mitigate the practical issues arising from an incomplete open set. Therefore, I will maintain my score.

**Key Questions For Authors:**

Questions:

1. How well does the proposed framework generalize to mixed categories and unseen semantic categories?

2. How does the current Router perform with unseen categories? Does it evenly distribute gating weights or make high-confidence erroneous routes to irrelevant experts?

3. Does "hard-sampling" refer to difficult samples?

4. Although the paper claims that the Mirage dataset solves the problem of outdated datasets, its construction details show that it is based on known knowledge of the generator. How can it guarantee coverage of current "in-the-wild" scenarios?

5. How are the training hyperparameters chosen?

6. Does the order of experts in the first stage affect performance? Because training expert i requires orthogonality with previous experts, this may limit the training of subsequent experts.

**Limitations:**

Please refer to "Strengths And Weaknesses" and "Key Questions For Author".

**Strengths And Weaknesses:**

Strengths:

1. The idea of ​​decoupling generated content and generation method is reasonable and intuitive.

2. A new dataset, Mirage, was constructed, which is of great significance to the AIGI detection community.

3. Experimental results show that the proposed strategy has good generalization performance.

Weaknesses:

1. The router relies on a limited number of predefined semantic categories, which may not cover all semantics. Furthermore, real-world images often contain overlapping semantic information. These concerns may cause the proposed framework to fail.

2. From the experimental results, although it has good generalization ability overall, its performance is not significant on some methods, and even worse than baseline methods. This requires further explanation, including whether it is related to the generation mechanism, which could help improve performance.

3. Whether the proposed Mirage dataset covers the wild as claimed in the paper, and the extent of its coverage, is unclear.

---

> ### Author Rebuttal · Authors · 2026-03-31
>
> Thank you for the valuable feedback on our paper.
> ### Q1. Limited predefined categories, mixed inputs, and unseen semantics
> We agree this is important. For mixed-category inputs, Top-K routing naturally enables multi-expert collaboration. As shown in Fig. 7 and 9, when an image contains both Animal and Human, the router assigns non-zero weights to both experts (e.g., 0.69 and 0.31), while the universal artifact expert remains active.
>
> For unseen/open-set semantics, we do not claim complete open-set coverage, and have acknowledged this limitation in the paper. Performance may degrade for categories far from predefined experts, but the always-active universal artifact expert (category-agnostic safety net) keeps degradation controlled. We validate this with 400 medical images absent from training (200 real, 200 fake), achieving 92% accuracy. The router distributions [Human: 0.37, Animal: 0.0, Object: 0.57, Scene: 0.06, Anime: 0.0] are semantically plausible: medical images depict organs, instruments, and scans (Object), with human anatomy activating the Human expert, showing the router gracefully leverages the most relevant existing experts for unseen categories. Moreover, the orthogonal residual design inherently supports incremental expansion: new experts can be added without full retraining.
>
> ### Q2. Why the method is not uniformly better on every subset
> Performance differences across subsets relate to generation mechanisms and benchmark bias. Some detectors specialize in specific artifact families (e.g., SD1-series), performing well on overlapping subsets but poorly under distribution shift. OmniAID instead targets balanced universalization. Compared to Effort, OmniAID shows consistent gains: +4.8% on GenImage (95.9% vs. 91.1%), +15.29% on Chameleon (77.35% vs. 62.06%), +8.07% on Mirage-Test (51.10% vs. 43.03%), and +45.36% with OmniAID-Mirage (88.39% vs. 43.03%). The advantage grows on harder, more diverse benchmarks, confirming OmniAID's strength in universalization. Where baselines lead on narrow subsets, this more likely reflects alignment with specific generator cues rather than stronger universality.
>
> ### Q3. Meaning of hard-sampling
> In our setting, "hard-sampling" does not refer to traditional hard example mining but to **an expert-specific data construction strategy** that suppresses shortcut cues, forcing each target expert to learn the specific type of evidence we intend it to capture.
>
> Specifically, for semantic experts, training uses only data from the corresponding semantic domain so the expert captures domain-specific semantic flaws. For the artifact expert, we use semantically-aligned real/reconstructed image pairs where semantics remain nearly invariant, forcing the model to rely on low-level artifact traces rather than exploitable semantic shortcuts. We will clarify this in the revision.
>
> ### Q4. Whether Mirage truly covers the wild.
> Mirage does not claim exhaustive coverage, but aims to better approximate in-the-wild scenarios. For Mirage-Train, beyond diverse generators and APIs, we additionally collect large-scale real-world synthetic images from online and third-party creation platforms. These uncontrolled, user-generated images with diverse prompts and post-processing help bridge the gap to in-the-wild data. For Mirage-Test, instead of only collecting visually realistic samples, we move to the source level by using realism-optimized SOTA generators specifically tuned for high fidelity and minimal artifacts, creating more challenging evaluation data. Thus, Mirage **significantly narrows the gap** to real-world AIGI scenarios from both data collection and generation perspectives.
>
> ### Q5. How training hyperparameters are chosen
> Hyperparameters are determined through systematic sensitivity analyses in Appendix E (Tables 13 to 19). For example, Top-K selection on Mirage shows the following trade-off:
> | Top-K | Throughput (FPS) | Mirage-Test | Δ Acc (vs. prev) | Δ FPS (vs. prev) |
> |:---:|:---:|:---:|:---:|:---:|
> | Top-1 | 201.38 | 87.01 | — | — |
> | **Top-2** | 191.99 | 88.39 | +1.38 | −4.7% |
> | Top-4 | 170.78 | 88.62 | +0.23 | −11.1% |
>
> Top-2 yields a 1.38% accuracy gain with only 4.7% throughput loss, while Top-4 adds marginal +0.23% at 11.1% throughput cost, making Top-2 the clear sweet spot.
>
> ### Q6. Whether expert training order affects performance
> We do not claim strict order-invariance. The theoretical analysis in Appendix B shows the orthogonality term is a **soft constraint regularizer**, not a hard mutual exclusion. Later-trained experts are encouraged to avoid redundant learning but can still capture complementary information. This effect is further moderated by the frozen backbone and coarse-grained partition. Experiments (trained on GenImage-SDv1.4) confirm order variations cause only acceptable differences:
> | Order | GenImage | Chameleon | Mirage-Test |
> |:---|:---:|:---:|:---:|
> | Current | **95.94** | 77.35 | **51.10** |
> | Reverse order | 95.75 | **78.28** | 48.49 |

---

> > ### Author Rebuttal · Reviewer_wGen · 2026-04-01
> >
> > Thank you to the authors for their response. Most of my concerns have been resolved, and I now have a deeper understanding of this work. Although the paper acknowledges the limitation that it cannot cover the complete open set, it represents a crucial step toward practical applicability, and it cannot be overlooked. Based on this consideration, I am maintaining my original score. However, I would not object to this paper being accepted.

---

> > > ### Author Response · Authors · 2026-04-01
> > >
> > > We sincerely thank the reviewer for the thoughtful follow-up.
> > >
> > > Regarding the concern about the limitation on complete open-set coverage, we would like to offer a more nuanced perspective for the reviewer and Area Chair to consider. We contend that, much like the construction of any large-scale dataset, achieving exhaustive coverage of all evolving visual semantics in "in-the-wild" scenarios remains a fundamental open problem in computer vision. Therefore, the hallmark of a truly practical system is not whether it offers a theoretically "perfect" solution to an infinite-domain problem, but how it **bounds performance degradation on unseen domains** and how it **enables low-cost scalability**.
> > >
> > > OmniAID addresses these requirements through three intrinsic architectural mechanisms:
> > >
> > > 1. **Coarse-grained Semantic Partitioning (Robust Generalization):** By training experts on broad, representative semantic clusters, our router effectively maps unseen categories to the most relevant learned features. Our zero-shot evaluation on **400 medical images** (entirely absent from training) yielded a **92% accuracy**. Crucially, the router's output for these images (e.g., **Object: 0.57, Human: 0.37**, etc.) reflects a semantically plausible decomposition, mapping medical scans and instruments to 'Object' and human anatomy to 'Human'. This proves that OmniAID does not experience functional collapse on unseen semantics but instead gracefully leverages related knowledge via its coarse-grained understanding.
> > >
> > > 2. **Universal Artifact Expert (The "Safety Net"):** In addition to semantic-specific experts, our framework incorporates a dedicated, category-agnostic artifact expert. This serves as a critical safety net to capture low-level generative traces regardless of high-level semantic content, ensuring robust detection even when the router encounters unfamiliar categories.
> > >
> > > 3. **Orthogonal Residual Design (Modular Scalability):** The practical value of OmniAID lies in its modularity. Unlike monolithic models that require exhaustive retraining for new domains, our orthogonal MoE architecture allows for the **incremental addition of new experts** without catastrophic forgetting. This "plug-and-play" capability provides a realistic, low-cost pathway to continually narrow the open-set gap as new generative threats emerge.
> > >
> > > In summary, we argue that OmniAID provides a scalable, decoupled framework that **inherently** confronts the open-set challenges of the real world. Based on these clarifications and the demonstrated robustness on out-of-distribution data, **we respectfully hope the reviewer will re-evaluate our work**. We believe the architectural progress presented here offers a viable path toward the "practical applicability" the reviewer highlighted. We will ensure these mechanisms are explicitly detailed in the final manuscript.

---

### Official Review · Reviewer_NFFT · 2026-03-13

**Soundness:** 3
**Presentation:** 3
**Significance:** 3
**Originality:** 3
**Overall Recommendation:** 4
**Confidence:** 5

**Summary:**

This paper proposes OmniAID, a Mixture-of-Experts based framework for universal AI-generated image detection that aims to improve generalization across different semantic domains and generative models. The key idea is to decouple content-dependent semantic flaws from content-agnostic generator artifacts by using multiple specialized semantic experts together with a fixed universal artifact expert, combined through a routing mechanism and a two-stage training strategy. In addition, the paper introduces a new dataset, Mirage, designed to better reflect modern generative models and real-world scenarios. Experiments on several benchmarks, including cross-domain and in-the-wild settings, show that the proposed method achieves improved robustness compared to existing detectors.

**Compliance With Llm Reviewing Policy:**

Affirmed.

**Final Justification:**

The author provided a detailed rebuttal, addressing my concerns.

**Key Questions For Authors:**

1.The paper claims that VFM-based detectors learn entangled representations mixing semantic flaws and low-level artifacts, but this seems inconsistent with prior work such as C2P-CLIP (AAAI 2025), which suggests CLIP can generalize via similarity-based matching without relying on low-level signals. Could the authors clarify how their observations align with these findings and provide more analysis supporting the entanglement assumption?

2.The paper introduces multiple semantic experts and one universal artifact expert, but the rationale for the expert division and the chosen number of experts is unclear. How sensitive is the method to the number of experts, and is there a principled way to determine the expert configuration?

3.The role of the Fixed Universal Artifact Expert is somewhat vague. It is not entirely clear how this expert is trained, how it differs from semantic experts, and why a single artifact expert is sufficient to capture all content-agnostic signals.

4.The MoE architecture is a key contribution, but the discussion of related MoE-based methods is limited.

**Limitations:**

yes

**Strengths And Weaknesses:**

**Strengths:**
The paper addresses an important and timely problem in AI-generated image detection and proposes a novel decoupled Mixture-of-Experts framework that separates semantic flaws from generator artifacts, which is an interesting and well-motivated design. The proposed two-stage training strategy and the introduction of the Mirage dataset further strengthen the work, and the experimental results are comprehensive, including cross-domain, cross-generator, and in-the-wild evaluations. Overall, the method shows strong empirical performance and provides useful insights into improving generalization for AIGI detection.

**Weaknesses:**
The paper lacks sufficient analysis in several key aspects. The assumption that existing VFM-based detectors learn entangled semantic and artifact features is not fully justified and appears to conflict with prior work such as C2P-CLIP. The design choices of the MoE framework, including the number of experts and the role of the universal artifact expert, are not thoroughly analyzed, and the related work discussion on MoE-based models is limited.

---

> ### Author Rebuttal · Authors · 2026-03-30
>
> Thank you for the valuable feedback on our paper.
>
> ### Q1. The relation between our claim and C2P-CLIP
> We thank the reviewer for raising C2P-CLIP, which in fact supports rather than contradicts our observation.
>
> C2P-CLIP shows that CLIP-based detectors rely on high-level semantic similarity rather than genuine forensic evidence. This aligns with our finding that current detectors learn semantically dominated representations, where artifact signals are secondary. Instead of tackling the fundamental challenge of artifact modeling, C2P-CLIP chooses to accommodate semantic reliance via category-specific prompts, representing a **pragmatic compromise** within the semantic-dominated paradigm. While effective in simple settings, it becomes fragile under in-the-wild shifts, as no artifact-level evidence is available when semantics fail. This is supported by Fig. 2a and 2b, where accuracy drops under semantic shift despite unchanged artifacts.  Fig. 6a (t-SNE) provides direct visual evidence of entanglement: rather than forming real/fake clusters based on forensic features, the embedding space organizes primarily by semantic category, with real and fake samples of the same category interleaved and inseparable, yielding an overall chaotic intermingling that leaves no discernible forensic boundary.
>
> This is precisely the gap OmniAID addresses. (1) On the semantic axis, C2P-CLIP uses external prompts in a shared space, while OmniAID performs intra-model decoupling via routable experts, yielding more structured representations. (2) We introduce the missing artifact axis. A fixed universal artifact expert acts as a content-agnostic forensic anchor, which is absent in C2P-CLIP.  Fig. 6b (t-SNE) effectively demonstrates clear separation between semantic content and artifacts. We will clarify this relationship in the revision.
>
> ### Q2. The division for semantic experts and their number
> Our partition follows broad categories used in in-the-wild evaluations (e.g., Chameleon): **Human, Animal, Object, Scene**, plus **Anime** (common in modern synthetic imagery yet absent from prior datasets). Proposition A.4 shows finer granularity theoretically improves decoupling, and experiments (trained on Mirage-Train) confirm consistent gains:
> | Number | Partition | GenImage | Chameleon | Mirage-Test |
> |:---:|:---|:---:|:---:|:---:|
> | 1 | {All} | 95.73 | 84.55 | 72.38 |
> | 2 | {Real}, {Anime} | 96.53 | 87.81 | 82.99 |
> | 3 | {Ani, Hum}, {Obj, Sce}, {Anime} | **97.24** | 89.76 | 86.82 |
> | 4 | {Ani}, {Hum}, {Obj, Sce}, {Anime} | 96.74 | 90.07 | 87.87 |
> | 5 | {Ani}, {Hum}, {Obj}, {Sce}, {Anime} | **97.24** | **91.42** | **88.39** |
>
> Performance jumps significantly when decoupling "Anime" from "Real-world" ($N_s=2$), with further gains as experts specialize in finer semantic splits such as Living vs. Non-living ($N_s=3$).
>
> Determining an optimal configuration is inherently a trade-off. We recommend three principled guidelines: (1) **semantic coverage**: span major real-world semantic domains; (2) **per-expert data sufficiency and balance**: each expert must have enough samples to specialize reliably (e.g., on GenImage-SDv1.4, certain categories are too scarce to train a reliable expert, so we adopt coarser groupings to ensure data adequacy); and (3) **tractable partition cost**: overly fine taxonomy increases annotation cost and may degrade the accuracy of manual or LLM-assisted annotation. Beyond these, automated partitioning via self-supervised semantic clustering (e.g., k-means on CLIP embeddings) is a highly promising future direction that we will discuss in the revision.
>
> ### Q3. The training and role of the fixed universal artifact expert
> The universal artifact expert is trained on semantically aligned real/fake pairs: real MS-COCO images and their VAE reconstructions (SD v1.x to SD3.5, TAESD, TAESDXL). Since pairs are semantically near-identical, the model cannot exploit content shortcuts and must learn low-level reconstruction inconsistencies. This differs from semantic experts which specialize in category-specific features.
>
> One expert suffices, as aligned training aims to capture shared artifact patterns across various generators, rather than all content-agnostic signals. It serves as a **stable cross-domain forensic anchor** providing always-available baseline signal regardless of semantic content. Table 5 confirms removing it causes the largest OOD drop; Table 12 shows routing it degrades performance, validating the fixed design.
>
> ### Q4. The discussion of related MoE-based methods
> We agree and will expand this. Recent MoE-based detectors use token-level routing (CLIPMoLE) or network-level adapter ensembles (Forensic-MoE) for multi-source aggregation. OmniAID differs in that routing dispatches at the image level by semantic category, experts operate within the parameter space of a single frozen model rather than as independent networks, and a fixed artifact expert bypasses routing entirely for content-agnostic forensic signals.

---

> > ### Author Rebuttal · Reviewer_NFFT · 2026-04-04
> >
> > The author provided a detailed rebuttal, addressing my concerns.

---

> > > ### Author Response · Authors · 2026-04-06
> > >
> > > We sincerely thank the reviewer for the acknowledgment and for the positive feedback on our rebuttal.
> > >
> > > We notice that the status updates to "(b) Partially resolved - I have follow-up questions for the authors" but we do not see any specific questions in the comment section. We kindly seek clarification on whether there are specific points or questions you would like us to address. If there are no further questions, we sincerely hope you might re-evaluate our work. We are more than happy to provide any further information needed.

---

### Decision · Program_Chairs · 2026-04-30

**Decision:**

Accept (regular)

**Comment:**

This paper proposes a decoupled Mixture-of-Experts framework for universal AI-generated image detection that separates semantic flaws across content domains from content-agnostic artifacts. It also introduces Mirage, a large-scale contemporary benchmark, a meaningful independent contribution.

All four reviewers acknowledged the motivation is intuitive, well-justified, and the empirical results are strong and consistent across benchmarks. Reviewers raised concerns that the authors addressed substantively.

Given the strong contribution, I recommend acceptance.